# Find your Needle: Small Object Image Retrieval via Multi-Object Attention Optimization

**Michael Green**[1*], **Matan Levy**[2*], **Issar Tzachor**[1*], **Dvir Samuel**[1,3], **Nir Darshan**[1], **Rami Ben-Ari**[1,§]

[1]OriginAI, Israel
[2]The Hebrew University of Jerusalem, Israel
[3]Bar-Ilan University, Israel

## Abstract

We address the challenge of Small Object Image Retrieval (SoIR), where the goal is to retrieve images containing a specific small object, in a cluttered scene. The key challenge in this setting is constructing a single image descriptor, for scalable and efficient search, that effectively represents all objects in the image. In this paper, we first analyze the limitations of existing methods on this challenging task and then introduce new benchmarks to support SoIR evaluation. Next, we introduce Multi-object Attention Optimization (MaO), a novel retrieval framework which incorporates a dedicated multi-object pre-training phase. This is followed by a refinement process that leverages attention-based feature extraction with object masks, integrating them into a single unified image descriptor. Our MaO approach significantly outperforms existing retrieval methods and strong baselines, achieving notable improvements in both zero-shot and lightweight multi-object fine-tuning. We hope this work will pave the way and inspire further research to enhance retrieval performance for this highly practical task. Project page: https://pihash2k.github.io/findyourneedle.github.io.

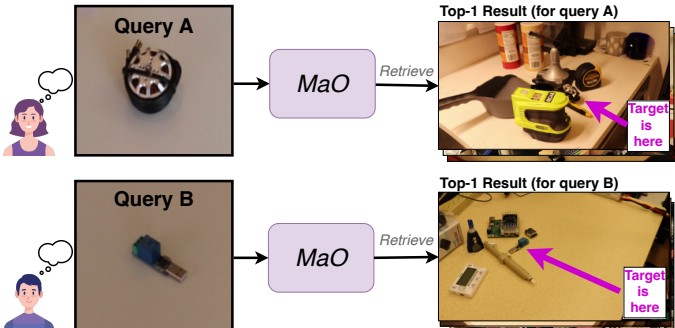

Figure 1: It is highly challenging to find an object instance that appears small in a cluttered scene image, moreover when shuffled in a large corpus of images. Top-1 retrieval results by our Multi-Object Attention Optimization (MaO) method are shown, attained with a single representation per-image, allowing a scalable search.

## 1 Introduction

Searching for a small object within a large image corpus is akin to finding a needle in a haystack. The challenge arises from the object's small size, the existence of many distracting objects in the

---

[*]Equal Contribution
[§]Corresponding author: <ramib@originai.co>

39th Conference on Neural Information Processing Systems (NeurIPS 2025).

scene, and moreover, the vast number of images in the corpus. In this paper, we address the problem of Small Object Image Retrieval (SoIR), where the goal is to search in a large image corpus and identify the images containing a specific (or personalized) instance of a likely small object, in a cluttered scene. In other words, the task is not just about locating any needle in the haystack but about finding *your* specific needle. In Image Retrieval (IR), a single global descriptor offers a strong advantage in large databases, due to computational and memory benefits. They provide a fast way to perform retrieval and allow additional speed-up with approximate nearest neighbor search techniques [36, 29, 14, 37, 46, 18].

SoIR can be seen as a specialized case of Instance-Based Image Retrieval (IBIR), where the objective is to retrieve images from a large corpus that contains the exact instance of a query object or scene [37, 50, 34, 35, 28]. Recent state-of-the-art (SoTa) IBIR models [37, 20, 2, 36, 11] have primarily been evaluated on datasets such as $\mathcal{R}$Paris6K, $\mathcal{R}$Oxford5K [27], Products-10K [3], and GLDv2 [48], which predominantly feature large, centrally positioned objects. However, retrieving images containing small objects in cluttered scenes remains largely unexplored. Recent findings [35] indicate that existing IBIR methods struggle in scenarios where multiple objects are present, particularly when datasets include similar objects from the same category. The INSTRE [47] dataset has been introduced as an additional benchmark for the IBIR task [11, 20, 26, 6]. While benchmarks such as $\mathcal{R}$Paris6K and $\mathcal{R}$Oxford5K contain objects occupying approximately 40% of the image area on average, INSTRE features smaller objects, covering 20–25% of the image, making retrieval more challenging. However, INSTRE images typically contain only 1–2 objects per image, limiting the complexity of multi-object interference. Recently, Li et al. [19] introduced the VoxDet dataset, originally designed for *object detection*. We find it particularly suitable for SoIR due to the small object sizes and the presence of many objects within each scene. Figure 1 illustrates these challenges with example images and retrieval results. For better evaluations of SoIR methods, we introduce three benchmarks based on VoxDet, INSTRE and PerMiR [35], highlighting the significant performance drop of existing IBIR methods in SoIR. Additionally, we propose a novel retrieval approach tailored to this task.

We define small and cluttered objects in the following contexts: (a) objects occupying a small fraction of the image area (*e.g.*, 0.5-1%), (b) cluttered scenes with multiple objects (often 10+), including same-category instances. The key challenges in retrieving small objects are: (1) capturing the object instance characteristics in the image representation, even when the object is small (2) ensuring that all objects in a scene are well-represented in a single image descriptor.

The representation of small objects in images has recently garnered notable interest in different contexts. Recent research in multi-modal large language models (MLLMs) [51], suggests that the primary challenge in object representation within images is ensuring the quality of its representation. Studies in object-level multi-modal learning [31] has explored these challenges within tasks such as text-to-image semantic search [16, 17, 6] and instance localization [7]. Additionally, Samuel et al. [35] demonstrated that the presence of multiple objects within an image can cause significant confusion and performance degradation in multi-object scenarios, while Abbasi et al. [1] highlighted a strong encoder bias toward larger objects, further complicating small object retrieval.

To address these challenges, we propose a novel SoIR method, as illustrated in Figure 2. Our method first decomposes the image into individual objects using an open-vocabulary detector (OVD), encodes them separately, and then integrates their representations into a single global descriptor. By isolating and encoding each object separately, our approach ensures balanced representation for all objects, regardless of size, while filtering out irrelevant background elements (*e.g.* sky, walls). Additionally, we refine the image representation through a post-training attention optimization process, aligning the image descriptor with object attention maps to improve object representation and eventually, retrieval accuracy. We refer to this method as Multi-object Attention Optimization (MaO).

We extensively evaluate our approach through comprehensive experiments, demonstrating its superior performance against existing methods and strong baselines. The results highlight our method's ability to effectively retrieve images containing small instances of a target object embedded in cluttered images. Sample results are shown in Fig. 1.

We summarize our contributions as follows:

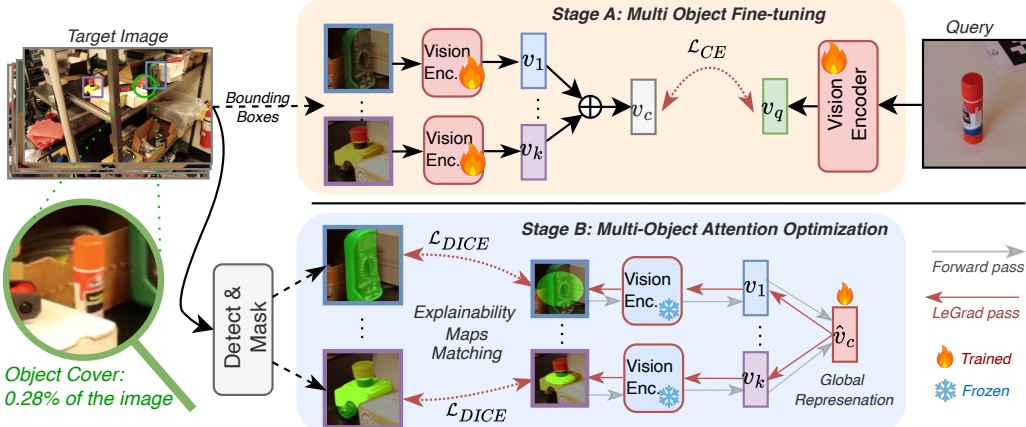

Figure 2: Overview of our MaO encoding method. **Stage A (Top):** Contrastive fine-tuning where for each training image, objects are cropped and encoded using a visual backbone. The objective is to align the average object-wise representation with a single object query. **Stage B (Bottom):** The global image representation is optimized by simultaneously aligning the explainability maps of object crops with their corresponding object masks, leveraging the pre-trained visual backbone from Stage A. Note that we optimize a single vector with respect to-K different image crops (K objects), while backpropagating gradients only through a single (unified) vector. Our global image descriptor, derived from MaO, enables efficient search for the occurrence of a specific small object, during inference.

- We study and analyze the problem of Small Object Image Retrieval (SoIR), identifying key challenges and limitations in existing methods. Our study focuses on three critical factors: object size, clutter (scenes with many objects), and resolution.

- We introduce new benchmarks specifically designed for SoIR, enabling more rigorous evaluation and fostering future research in this domain.

- We propose MaO, a novel method that leverages explainability maps to refine object representations and unify them into a single image descriptor. Our approach outperforms existing methods and strong baselines.

## 2 Related Work

Below, we review key approaches relevant to our work.

**Instance-Based Image Retrieval:** CNN-based methods have played a fundamental role in IBIR for many years. Region-based techniques [44, 32, 6] explore various region-matching strategies to compute image similarity. For instance, GeM [29] enhances retrieval performance by fine-tuning a CNN network, introducing the Generalized-Mean (GeM) pooling layer. Despite their effectiveness, these methods are computationally expensive and rely on predefined, grid-based regions, posing two major drawbacks: (1) the predefined grid is content-agnostic, making it unlikely to align with object boundaries, and (2) many grid regions capture irrelevant background information, creating a clutter in the image representation. Moreover, these approaches tend to emphasize larger objects while failing to adequately represent smaller ones. In contrast, we replace rigid grid sampling with an open-vocabulary object detector (OVD) that detects arbitrary objects, allowing for a more flexible and content-aware representation. Each object representation is further refined through attention-based encoding, allowing the integration of multiple objects into a single image descriptor.

**Local feature based methods:** A different approach suggests to represent an image with a set of local features while using a similarity function to compare two local descriptor sets [14, 38, 15, 42]. Due to high computational cost, these methods are frequently used for small scale datasets or top-K re-ranking and are not suitable for large scale image retrieval. Some methods attempt to make local feature based methods more efficient *e.g.* [14, 37, 38]. Other methods such as GSS [20] and HP [2] leverage graph networks to perform spatial verification. However, these approaches require storing

multiple local features for each gallery image or performing additional per-query computations, beyond simple vector similarity, leading to increased memory consumption and computational overhead. In contrast, our approach produces a single compact representation per image, ensuring both memory efficiency and fast search capability across large datasets. SuperGlobal [37] similarly focuses on memory-efficient retrieval with a re-ranking stage based solely on feature refinement. However, its performance drops significantly when encountering images with small objects.

While CNNs have traditionally been the dominant approach for IBIR, PDM [35] was among the first to leverage foundation models for this task. Despite its strong performance in cluttered scenes, PDM relies on large feature maps and a prompt for pairwise comparisons, making it less suitable for only an image based query and scalable global search.

**Detection-based IBIR:** Previous works have explored generating a global representation by aggregating region-wise descriptors. Gordo et al. [9] employs a region proposal network to learn which regions should be pooled to form the global descriptor, while Sun et al. [39] uses a generic object proposal to identify regions and aggregating their SIFT [21] features into a compact representation. Recent work [43] combines a custom domain-specific detector with a CNN backbone and a match-kernel-based aggregation [45]. In contrast to these methods, which depend on custom detectors or keypoint-based features, our approach fine-tunes a visual encoder for the multi-object representation task. We further refine object-specific representations by leveraging object masks, while employing an effective aggregation strategy to enhance global representation.

**Small Objects**: While IBIR methods have primarily focused on benchmarks where images contain a single, prominent object, few studies have specifically addressed the retrieval of small objects [20, 11, 6], mainly evaluating performance on the INSTRE dataset [47]. To better assess small object retrieval, we introduce two new challenging subsets from INSTRE, named INSTRE-XS and INSTRE-XXS. Additionally, we establish a new benchmark using the VoxDet dataset [19], dubbed *VoxDet-SoIR*, which features small objects in cluttered scenes, a scenario that has not been explored in prior instance-based retrieval research.

**Object-Focused Representation:** $\alpha$-CLIP [40] introduced a CLIP variant with an additional channel for object masks, enabling object-focused representations. As we demonstrate in this study, $\alpha$-CLIP provides a more effective solution for handling small objects compared to CLIP and some previous methods. In this work, we show how existing foundation models like CLIP and DINOv2 can be leveraged for encoding an image containing small objects. We effectively utilize object masks to improve the representation of small objects and enhance retrieval of images with the target instances, even in cluttered scenes.

Recently, Bousselham *et al.* [5] introduced MaskInversion, a method used for referring expression classification and localized captioning. It generates an object-focused representation, by optimizing an image token to align with the image's explainability map. Inspired by this approach, we propose a novel feature refinement stage in our framework. However, instead of operating on a single image, our method integrates multi-object attention from multiple image crops into a unified global representation. This refinement process is guided by our image decomposition and multi-object fine-tuning strategy, ensuring a more comprehensive and scalable retrieval solution.

## 3    Method

Our task is to retrieve all images from a large gallery that contain the object depicted in a given query image. In this section, we present Multi-object Attention Optimization (MaO), a method that encodes both query and gallery images while tackling three key challenges: (1) the small size of target objects within images (2) clutter: the presence of multiple distracting elements in gallery images, existing in the foreground and background (3) Deriving a single image representation for each gallery image. Instead of encoding entire images as a whole, our approach first isolates individual objects, encodes them separately, and then refines their encoding by finding an optimized representation for all objects in the scene with a single descriptor. This structured encoding reduces the impact of irrelevant regions (*e.g*. sky, ceiling, or wall) while preserving essential object-level details. Our method consists of two stages: (A) Multi-Object fine-tuning of a visual backbone to learn a representation for multiple objects within an image, and (B) post-training refinement, where the initial object representations are jointly refined and unified through representation optimization, leveraging object masks and attention-based cues. A schematic overview of our method is presented in Figure 2.

**Feature Extraction:** First, we extract and filter relevant image information for the task, focusing on individual objects. Let us consider an RGB image $X \in \mathbb{R}^{3 \times W \times H}$. Using given annotations or an open-vocabulary object detector (OVD), we divide the image into $k$ crops, $x_i \subseteq X$, for $i = 1, 2, \ldots., k$, each centered on a single object. To this end, we leverage the OVD for object-proposal (agnostic to the object category). Each object is then encoded using a visual backbone, producing $k$ feature vectors: $\{v_1, \ldots, v_k\} \in \mathbb{R}^d$. This process filters irrelevant regions, improving feature focus for the SoIR task.

**Multiple Object Fine Tuning (Stage A):** Our objective is to preliminary fine-tune our backbone on a multi-object and instance based scenario. To achieve this, we align each query image with a representation computed from its corresponding ground truth gallery image. For a query image (an object-focused image), we extract its feature vector $v_q$ using the visual encoder backbone. For the corresponding gallery image, we fuse multiple object representations by average pooling the $k$ object-level features, yielding $v_c$ (see Figure 2-Stage A). We then compute the cosine similarity between $v_c$ and $v_q$, and train the model using the InfoNCE contrastive loss [24] over the batch. This process is schematically described in Figure 2-Stage A. The Multi-Object Attention Optimization stage is applied post-training, producing the final refined representation $\hat{v}_c$ as described below.

**Multi-Object Attention Optimization (Stage B):** LeGrad [4], introduced a gradient-based method for explainability in ViT that relies solely on attention map gradients. We leverage LeGrad [4] to optimize a *single* token (representation) based on the image's attention map. However, instead of applying a single binary mask to a single image, we refine the token by aligning it with $k$ different images. Each crop's explainability map is optimized to align with its corresponding mask, ensuring that the final token effectively captures the object within each crop. Essentially, we optimize a single vector with respect to-$k$ different image crops ($k$ objects), while backpropagating gradients only through a single (unified) vector (as illustrated in Fig.2-Stage B). This process ensures that the final vector effectively represents all objects in the image, while maintaining a compact yet expressive feature encoding. The attention-based representation suggests an advantage, as it effectively encodes small objects even when they are not perfectly centered in the bounding box. For the query image, the same optimization is applied, while considering a single object.

For each crop, we assume the availability of an object mask, $m_i \in \mathbb{R}^{W_i \times H_i}$, typically obtained using SAM [12]. Leveraging our trained vision encoder (from stage A), we first encode the image crops $x_i$ to obtain $v_i$. Using LeGrad [4], we compute an explainability (heat) map, $E(v_c \cdot v_i) \in \mathbb{R}^{W_i \times H_i}$, indicating the attention map associated with a given representation vector $v_i$. To attain a single representation, we then optimize a *single token* $\hat{v}_c$ for an explainability map of *all objects* in the image (within individual crops). The objective of this optimized token is defined as:

$$\hat{v}_c = \operatorname*{argmax}_{v_c} \sum_i \operatorname{IoU}\left(E(v_c \cdot v_i), m_i\right) + \alpha\, v_c \sum_i v_i \tag{1}$$

This optimization aims to adjust the token's explainability map to best align with all object masks. Here, IoU denotes the Intersection over Union measure between the explainability map and object mask, while $\cdot$ represents the normalized dot product (*i.e.*, cosine similarity). For preliminary details on LeGrad and computing of $E$ please refer to Appendix A. During optimization we initialize $\hat{v}_c$ with the $v_c$ obtained from stage A, $\hat{v}_c^{(0)} = v_c$. Following [5], we also incorporate a regularization term controlled by $\alpha$ to ensure that the optimized token remains close to its initial embedding, $v_c$. Note that the new representation obtained from Stage B, is based on all object embeddings derived from our stage A trained encoder. The optimization process is performed using standard gradient descent, resulting in a refined representation vector for each gallery image, denoted as $\hat{v}_c$. Image retrieval is then conducted via a conventional, vectorized similarity search in the gallery.

Our experiments show that the proposed object-based aggregation model, which extends Mask-Inversion to handle multiple objects, provides strong benefits for small-object retrieval, a challenge that is often underexplored by existing methods. An additional advantage arises from applying Stage-B refinement to the customized multi-object encoder trained in Stage-A, resulting in two tightly integrated and complementary stages.

## 4   Evaluation

In this section, we conduct extensive experiments across multiple datasets to assess our method's performance on the Small Object Retrieval (SoIR) task. We compare our approach against several

methods, demonstrating that: (1) state-of-the-art IR methods suffer a significant performance drop in the SoIR task, and (2) our proposed method, MaO, substantially improves retrieval performance, particularly for small objects in the gallery and cluttered scenes. As baselines, we first evaluate conventional IR methods that encode the entire image into a single global representation using foundation visual encoders such as CLIP [30], DINOv2 [25], and $\alpha$-CLIP [40]. Additionally, we compare against previous IBIR methods, including GSS [20], GeM [29], CVNet [14] and SoTA SuperGlobal (SG) [37] and AMES [38], using each with its corresponding global and re-ranking component, if exists, and as prescribed, *e.g.* in SG (with CVNet) and AMES (with DINOv2 and SG pre-compute [38]). We also fine-tune DINOv2, CLIP, and GeM on VoxDet and assess their performance on all of our benchmarks. We evaluate MaO using the widely adopted mean average precision (mAP) metric. For all models we use the public pre-trained weights and keep the original setting.

**Implementation Details**: We use the *AdamW* optimizer, initializing the learning rate at $5 \times 10^{-5}$ with an exponential decay rate of $0.93$ down to $1 \times 10^{-6}$. We fine-tune all the transformer-based models on the VoxDet training set using a LoRA [10] adapter of rank 256. We train with a batch size of 128 for 1 epoch, across four *NVIDIA-A100* nodes. For inference, we apply OWLv2 [22] as OVD, applied in object-proposal mode, to detect any object in gallery images, considering bounding boxes with a confidence threshold above $0.2$. For detected objects, we enforce a minimum bounding box size equal to the backbone's input dimensions by centering a cropped region around the object. The refinement process is done for 80 iterations with $\alpha = 0.03$ and a learning rate of $1 \times 10^{-1}$, which takes $0.03$ seconds of computation per object. For gallery images, this is performed offline. Our global feature dimension is the same as the backbone, 512D for CLIP and 768D for DINOv2.

## 4.1 Datasets

Creating instance-based datasets is particularly challenging, as it requires the same object instance to appear across multiple images in varying positions, backgrounds, and configurations. Most commonly used IBIR datasets focus on large, prominent objects, such as $\mathcal{R}$Paris6K, $\mathcal{R}$Oxford5K [27], and Google Landmark Dataset (GLDv2) [48], or object-centric datasets like Products-10K [3] where objects are typically large and centered in the image (see Table 1). We define *Object Size* as the ratio of the object-mask area to the total image area. Table 1 presents fundamental characteristics of several datasets used for IBIR task, in terms of clutter and object size.

**INSTRE**: The INSTRE dataset [47] presents a more challenging setting by including 1–2 smaller objects per image. It comprises 28,543 images across 250 object classes and provides bounding box annotations. To adapt it for our evaluation, we introduce two new subsets: **INSTRE-XS** (Extra Small): Containing 2,428 queries and a gallery of 2,065 images, with objects occupying less than 15% of the image area. **INSTRE-XXS** (Extra-Extra Small): A more challenging subset with 106 queries and a gallery of 120 images, where objects are at most 5% of the image.

**PerMiR**: The PerMiR dataset [35] focuses on multiple object instances across 16 categories (*e.g.* cars, people, animals, and food items). It comprises 150 queries and a gallery of 450 images, posing a greater challenge due to the presence of multiple instances per image, often including objects from the same category (but not the same instance) as the query. PerMiR also provides object mask annotations, which enable object-level evaluation.

**VoxDet-SoIR**: Recently, Li et al. [19] introduced VoxDet, the largest instance-based dataset designed for object detection, featuring 3D voxel-rendered images. The synthetic training set contains 9.6K instances, featuring 55K scenes and 180K bounding boxes, and the real test set includes 20 unique instances, $9,109$ annotations, and 24 complex, cluttered scenes with diverse poses, lighting, and shadows. For evaluation, we converted VoxDet into a large-scale instance-based retrieval dataset, name it as *VoxDet-SoIR* (VoxDet for Small Object Image Retrieval). The training set consists of distinct objects that do not overlap with those in the test set, averaging 6.34 objects per gallery image with an average object size of 2.82%. This design ensures the dataset's suitability for both multi-object and small-object scenarios. For the rest of the paper we'll refer to this dataset as **VoxDet** for sake of brevity.

To summarize, we present 4 benchmark datasets for SoIR, each addressing a distinct retrieval challenge: (1) INSTRE-XS/XXS: Small objects, but with minimal clutter (2) PerMiR: Instance based retrieval in a cluttered scene, with less emphasis on object size. (3) VoxDet: Both small and

multi-object retrieval. Table 1 compares key properties of several benchmarks: number of Annotated (Annot.) or Detected (OVD) objects and their average size ratio. Our proposed benchmarks are highlighted in bold. Notably, VoxDet exhibits the most challenging scenario introducing the highest object density and the smallest object sizes.

For examples, more details and statistics see the Appendix. For a more detailed analysis, we created two versions of the VoxDet and PerMiR datasets. The first version, based on ground-truth (GT) object annotations, separates the impact of detection performance. The second version, referred to as the "in-the-wild" (VoxDet$_\mathcal{W}$, PerMiR$_\mathcal{W}$), uses object an OVD, particularly OWLv2 [22], simulating a more realistic, unconstrained retrieval scenario. Since INSTRE inherently contains only a single annotated object per image, we evaluate it exclusively in the "in-the-wild" setting. Detector performance details can be found in the Appendix.

Table 1: Benchmarks: Key Properties.

|  | #Obj Annot. | #Obj OVD | Obj. Size (in %) |
|---|---|---|---|
| **VoxDet** | 5.8 | 14.7 | 1.1 |
| **PerMiR** | 4.7 | 10.4 | 13.3 |
| **INSTRE-XS** | 1 | 1.8 | 6.6 |
| **INSTRE-XXS** | 1 | 1.9 | 2.2 |
| INSTRE ($\mathcal{S}1$) | 1 | 1.8 | 21.0 |
| Products-10K | 1 | 2.1 | 27.1 |
| $\mathcal{R}$Oxford | 1 | 5.9 | 37.6 |
| $\mathcal{R}$Paris6K | 1 | 4.9 | 41.4 |

Table 2: Performance comparison (in mAP). All results are for a single global feature vector per image. We apply our MaO method to CLIP and DINOv2 backbones, in zero-shot and fine-tuning settings. As observed, MaO significantly improves performance across all backbones and benchmarks. Note that baselines that encodes the entire image yields identical results on VoxDet and VoxDet$_\mathcal{W}$ and PerMiR/PerMiR$_\mathcal{W}$. $\dagger$ indicates results with re-ranker.

|  |  | VoxDet | PerMiR | VoxDet$_\mathcal{W}$ | PerMiR$_\mathcal{W}$ | INSTRE-XS | INSTRE-XXS |
|---|---|---|---|---|---|---|---|
| Zero Shot | GSS [20] | 52.01 | 26.73 | 52.01 | 26.73 | 82.34 | 67.98 |
|  | CVNet$^\dagger$ [14] | 43.77 | 24.72 | 43.77 | 24.72 | 43.46 | 38.72 |
|  | SuperGlobal$^\dagger$ [37] | 47.33 | 17.48 | 47.33 | 17.48 | 56.11 | 33.02 |
|  | AMES$^\dagger$ [38] | 48.68 | 29.72 | 48.68 | 29.72 | 78.61 | 68.08 |
|  | GeM [29] | 51.08 | 25.98 | 51.08 | 25.98 | 74.74 | 53.27 |
|  | CLIP | 44.52 | 26.98 | 44.52 | 26.98 | 51.97 | 37.49 |
|  | DINOv2 | 51.23 | 40.57 | 51.23 | 40.57 | 38.92 | 31.08 |
|  | $\alpha$-CLIP | 45.30 | 88.21 | 42.06 | 28.96 | 33.77 | 16.44 |
|  | **MaO-CLIP** | 65.22 | 89.51 | 52.09 | 35.81 | **89.39** | **71.23** |
|  | **MaO-DINOv2** | **70.20** | **89.86** | **59.60** | **43.46** | 71.28 | 53.13 |
| Fine Tune | GeM | 61.45 | 41.20 | 61.45 | **41.20** | 82.61 | 65.58 |
|  | CLIP | 52.80 | 38.49 | 52.80 | 38.49 | 72.90 | 62.04 |
|  | DINOv2 | 54.33 | 30.47 | 54.33 | 30.47 | 53.39 | 44.08 |
|  | $\alpha$-CLIP | 59.74 | 90.13 | 44.22 | 38.37 | 60.18 | 29.48 |
|  | **MaO-CLIP** | 79.86 | **90.86** | 63.18 | 39.10 | **91.29** | **77.46** |
|  | **MaO-DINOv2** | **83.70** | 90.07 | **68.54** | 40.44 | 90.01 | 75.91 |

## 4.2 Results

Table 2 presents mAP results across several benchmarks. Among the top-performing methods, traditional instance retrieval techniques such as GSS[20] and GeM[29] often outperform foundation models. In the zero-shot test and the controlled setting (two left-most columns), *e.g.* on VoxDet, GSS achieves 52.01% mAP against 44.52% and 51.23% for CLIP and DINOv2 respectively. This result highlights also the limitations of local feature matching used in CVNet[14] and AMES[38], to effectively match small objects, especially in cluttered scenes. However, by applying our attention optimization approach (Stage B in Figure 2) to CLIP and DINOv2, we achieve significant performance boost, reaching 65.22% for CLIP and 70.20% for DINOv2, without any fine-tuning. MaO surpasses previous instance-based retrieval techniques by 18–26 mAP points on VoxDet. On PerMiR, mAP is boosted from a range of 17.48-40.57% for techniques without object masks and 88.21% on $\alpha$-CLIP (which incorporates object masks) to 89.86%. Furthermore, after fine-tuning on the VoxDet (synthetic) training set, all methods demonstrate improved performance. Notably, $\alpha$-CLIP exhibits a substantial performance gain, attributed to its effective use of object masks. Importantly, MaO, utilizing a multi-object fine-tuning strategy and mask optimization, consistently outperforms all other methods across both benchmarks.

Next, we present the results for the in-the-wild benchmarks: VoxDet$_\mathcal{W}$, PerMiR$_\mathcal{W}$, INSTRE-XS, and INSTRE-XXS, where an OVD [22] is employed. For global methods, the results on VoxDet and VoxDet$_\mathcal{W}$ remain identical since the entire image is processed (with no bounding boxes). MaO-CLIP and MaO-DINOv2 consistently outperform all other zero-shot methods, highlighting the robustness of our approach even in real-world scenarios. Fine-tuning further improves performance across all models, with MaO maintaining its leading position. The only exception occurs with DINOv2 on PerMiR$_\mathcal{W}$, where performance declines. We attribute this drop to DINOv2 over-adapting to the VoxDet domain, and increasing the domain gap, whereas PerMiR primarily contains objects such as people and vehicles. Although GeM CNN backbone shows strong improvement after full fine-tuning, instead of our finetuned model, MaO-DINOv2 remains the top-performing method on PerMiR$_\mathcal{W}$, in zero-shot. On INSTRE, all methods suffer performance drops when dealing with smaller objects, as evidenced by the difference between INSTRE-XXS and INSTRE-XS. However, MaO still achieves the best results with a significant margin on both datasets.

The results emphasize the robustness of MaO in handling small and multi-object scenarios with high clutter. Despite processing an average of nearly 15 detected objects per image, MaO consistently shows improved retrieval results with small target objects. This highlights that object characteristics are better captured in our global image representation.

Figure 1 presents qualitative examples demonstrating the effectiveness of our MaO approach in retrieving images containing extremely small objects, even in cluttered scenes. Additionally, Figure 4 compares Top-1 retrieval results under challenging conditions, contrasting MaO-CLIP with fine-tuned versions of CLIP and DINOv2 as well as GeM.

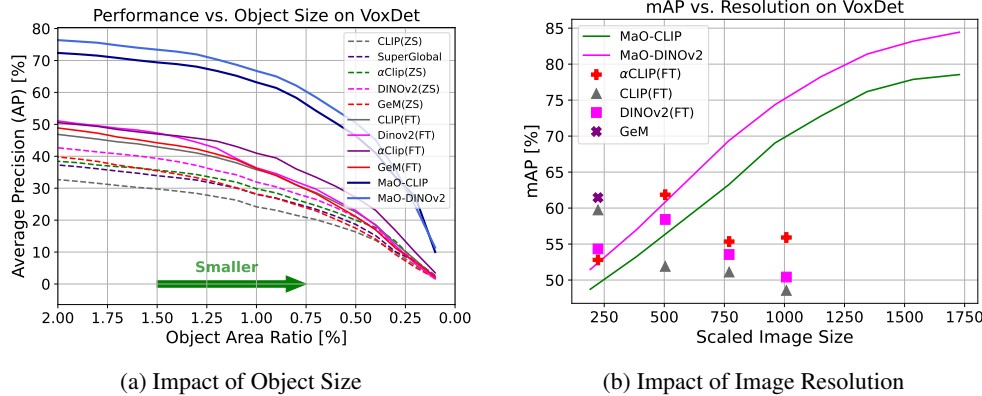

(a) Impact of Object Size          (b) Impact of Image Resolution

Figure 3: Retrieval performance analysis on our VoxDet benchmark. The results demonstrate key challenges in small object retrieval, showing the impact of object size (a) and image resolution (b) on retrieval accuracy. Performance declines as object size decreases. Higher image resolutions enhance retrieval effectiveness for MaO, whereas other alternatives marginally benefit from increased resolution. MaO consistently outperforms existing methods, showing robustness across all conditions.

Figure 5 further provides a visualization of explainability maps, comparing fine-tuned DINOv2 and MaO. While DINOv2's attention predominantly focuses on background elements, such as shelves, the attention maps of MaO are more effectively distributed across the relevant objects, highlighting the advantage of our Stage-B mask optimization, ensuring a more precise and object-focused representation. We also report in Appendix C the performance of MaO on non-small objects using PODS [41] and ILIAS [13] benchmarks. Appendix D presents a comparison with an MLLM method.

## 4.3 Controlled Analysis

Figure 3a shows the retrieval performance on VoxDet (measured in AP) as a function of the object size in the gallery, quantified by the mean object-to-image area ratio. The evaluation covers objects occupying roughly 0.1% to 2% of the total image area.

To isolate the impact of object size from detection quality, this experiment runs in a controlled setting, utilizing ground-truth (GT) object bounding boxes and object masks in both $\alpha$-CLIP and MaO. We observe that global fine-tuning consistently improves performance across all methods, as

indicated by the contrast between dashed and solid lines of the same color. Notably, MaO achieves the most significant performance gains with both CLIP and DINOv2 backbones, outperforming other approaches, even for very small objects (down to 0.5% of the image area). MaO surpasses previous retrieval techniques, including SuperGlobal [37], GeM [29], and fine-tuned models such as CLIP(FT), DINOv2(FT), and GeM(FT). Particularly, our method achieves an AP of 50% for objects occupying just 0.5% of the image area, demonstrating a substantial improvement over prior methods. The fine-tuning benefits of MaO are particularly pronounced compared to other techniques, reinforcing its effectiveness in handling small object retrieval in cluttered scenes.

Finally, we analyze the impact of image resolution on retrieval performance. Figure 3b illustrates how performance changes as images are resized to different resolutions, while the relative object size, remains constant. For global methods, retrieval performance either plateaus or declines as resolution increases. While $\alpha$-CLIP and DINOv2 exhibit slight improvements with higher resolution input, further increase in resolution leads to a drop in performance. This trend is likely due to the limitations discussed in [8], which attributes the decline to the "over-fitting" of foundation and pre-trained models on low-resolution training images. In contrast, MaO effectively leverages high-resolution images, consistently outperforming global methods. MaO further exhibits a predictable and gradual decline in performance with resolution decrease, demonstrating its robustness across different image scales.

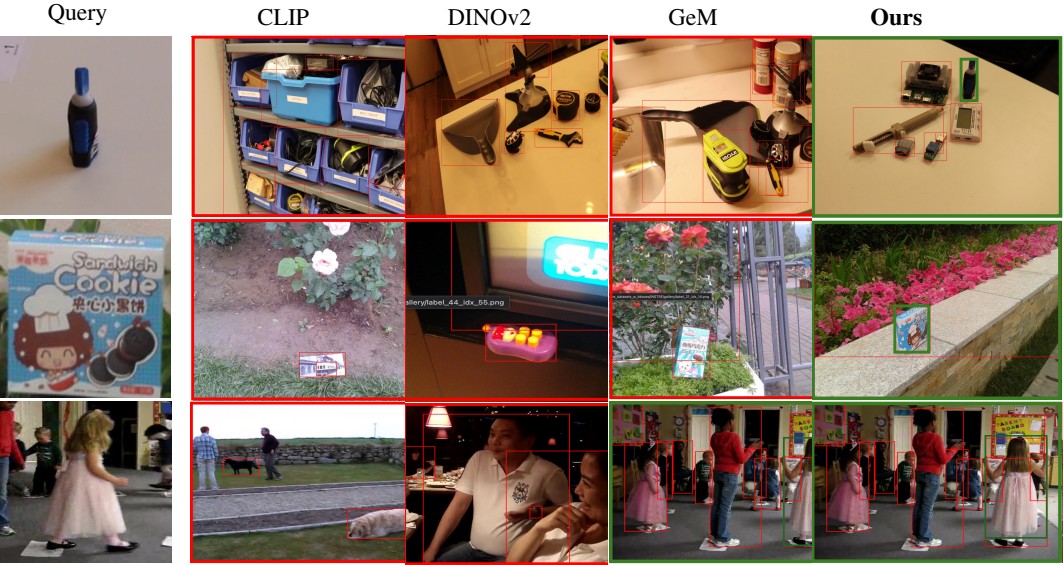

Figure 4: Examples of Top-1 retrieval results from VoxDet$_\mathcal{W}$ (top row), INSTRE-XXS (middle row), and PerMiR$_\mathcal{W}$ (bottom row). Red and green frames indicate incorrect and correct retrievals, respectively. In the images, red bounding boxes correspond to OVD detections, while a green bounding box highlights the target object. MaO successfully retrieves the correct image, even in cases with high levels of clutter and small objects.

## 4.4 Runtime Analysis

In this section, we report and compare the encoding and retrieval times for various methods. Most state-of-the-art retrieval methods adopt either a two-stage design with re-ranking (*e.g.* SuperGlobal) or rely on computationally intensive local matching (*e.g.* AMES), making them significantly less efficient during online search. We present a runtime analysis comparing MaO against two SoTA methods of SuperGlobal (SG) and AMES. Table 3 presents the runtime results, for both the encoding and the re-ranking stages (used in SG and AMES). All experiments were conducted on the VoxDet-SoIR benchmark, containing 1,581 images in gallery and 2,000 query images. The retrieval time based on vectorized (global) feature ranking is approximately 0.03 ms across all methods and is included in the search time.

The online search-time advantage of MaO is clear: at test time (*i.e.*, from receiving a query image to producing ranked retrieval results), MaO requires 30 ms for query encoding (single object). In comparison: SG takes 65 ms for query encoding and 0.37 ms for re-ranking (per-query). AMES takes

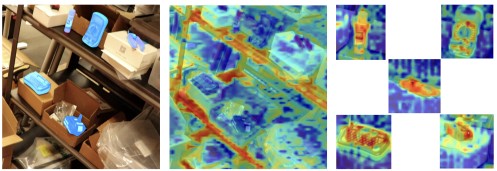

Figure 5: Explainability maps for the full image (center) and MaO (right) with DINOv2. Warmer colors indicate higher attention.

Table 4: Ablation study of our method on VoxDet and VoxDet$_\mathcal{W}$ (mAP).

| Method | VoxDet | VoxDet$_\mathcal{W}$ |
|---|---|---|
| Backbone ZS DINOv2 | 51.23 | 51.23 |
| + Fine-tuning | 54.33 | 54.33 |
| + Full image optimization | 69.54 | 48.24 |
| + Multi-object fine-tuning (Stage A) | 75.04 | 63.14 |
| + Multi-object optimization (Stage B) | **83.70** | **68.54** |

65 ms for query encoding (same as SG) but 2,286 ms for retrieval (per-query) due to the extremely heavy local matching process. Considering 2,000 queries the total search time is 0.8, 4572 and 0.06 sec for SG, AMES and MaO respectively. For offline gallery encoding, SG and AMES each require about 65 ms per image, while MaO takes 174 ms per image plus roughly 0.5 s per image for OVD pre-processing. Notably, we did not apply any acceleration techniques, such as batch processing or optimized OVD implementations (e.g., ONNX) for the OVD pre-processing. Consequently, our method delivers substantially higher accuracy (see Tab. 2) and is significantly faster than SG and AMES at test time, albeit with a greater offline encoding cost. Encoding time can be further reduced through code-level optimizations and parallelization, as the MaO pipeline is inherently well-suited for data parallelism.

Table 3: Runtime comparison on VoxDet dataset. Encoding time corresponds to an average of $\sim$ 5 objects per image. Search time corresponds to the retrieval stage for 2,000 queries, including vectorized search, re-ranking, and local matching where applicable. *: without OVD.

| Method | Encoding (ms/img) | Re-Ranking (ms/query) | Search Time (sec) |
|---|---|---|---|
| SG | 65 | 0.37 | 0.8 |
| AMES | 65 | $\approx 2286$ | 4572 |
| MaO | 174* | – | 0.06 |

## 4.5 Ablation Study

To further assess the contribution of each component in our method, we conduct an ablation study, presented in Table 4, which reports mAP results on VoxDet and VoxDet$_\mathcal{W}$ across different configurations. The baseline model is off-the-shelf DINOv2 with a global image representation. *Fine-tuning* is performed with VoxDet training set. *Full image optimization* extends the fine-tuned model by naively incorporating a global and single optimized attention map for all objects. This method in fact results in a performance drop on VoxDet$_\mathcal{W}$. *Multi-object fine-tuning* is MaO's stage A, and our *Multi-object optimization* (based on optimization for individual crops and aggregation) indicates our complete object-wise MaO method, which presents significant gains, on both benchmarks, showing the further impact for our masked optimization refinement. For more ablation studies including clutter impact please refer to the Appendix.

## 5 Summary and Discussion

While most prior IIR work examines large objects or scenes, the domain of retrieving small instances in cluttered environments, remains underexplored. To bridge this gap, we first establish benchmarks that enable an in-depth investigation of this challenge, highlighting the significant impact of object size and scene clutter on retrieval performance. We further propose MaO, a novel approach for SoIR that leverages foundation models such as CLIP and DINOv2. MaO encodes images in two stages: a multi-object fine-tuning stage, followed by a post-training refinement process (multi-object attention optimization) that yields an object-focused representation, ultimately generating a single-image descriptor for search and retrieval.

MaO achieves strong retrieval performance, successfully identifying objects as small as 0.5% of the image area. Nonetheless, certain limitations remain. Although prior work has demonstrated the effectiveness of detector-based retrieval methods (e.g., [9, 39, 43]), our performance can be affected by imperfect detector recall, resulting in slight retrieval drops. Appendix F presents a thorough analysis of the OVD impact and discusses further limitations of our approach, yet highlighting strong generalization capabilities. We hope this work will inspire further research in the challenging task of small object retrieval.

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

# Appendix

This Supplemental Material presents the following additional details concerning the experimental results and their analysis.

## A Preliminary on LeGraD and Explainability Map

**LeGrad** [4] is a gradient-based feature attribution method specifically developed for Vision Transformer (ViT) architectures. Unlike traditional explainability techniques such as Grad-CAM, which are optimized for convolutional networks, LeGrad exploits the attention mechanisms inherent in ViTs to generate an explainability map that highlights the most influential image regions contributing to the model's decision.

**Mathematical Formulation** Consider a vision-language model $F$ that processes an input image $x \in \mathbb{R}^{3 \times W \times H}$ and outputs an activation score $s \in \mathbb{R}$, which may represent a classifier's confidence level or the cosine similarity between vision and text embeddings. In a ViT with $L$ layers, let $A^l$ represent the attention maps at layer $l$. The gradient of the activation $s$ with respect to the attention maps is computed as:

$$\nabla A^l = \frac{\partial s}{\partial A^l}. \tag{2}$$

LeGrad determines the explainability score for each attention map $A^l$ by applying the ReLU function to eliminate negative contributions:

$$\hat{E}^l(s) = \frac{1}{hn} \sum_h \sum_i \text{ReLU}\left(\frac{\partial s}{\partial A^l_{h,i,.}}\right), \tag{3}$$

where $h$ denotes the number of attention heads and $n$ represents the number of visual tokens.

The final explainability map $E$ is obtained by averaging the scores across all layers:

$$\bar{E} = \frac{1}{L} \sum_l \hat{E}^l(s). \tag{4}$$

To construct the final explainability map, the influence of patch tokens is isolated, reshaped into a 2D spatial representation, and normalized using min-max scaling:

$$E = \text{norm}(\text{reshape}(\bar{E})). \tag{5}$$

## B Visualization of Explainability Maps

Figure B1 shows the explainability maps before and after our multi-object optimization stage on a sample from VoxDet.

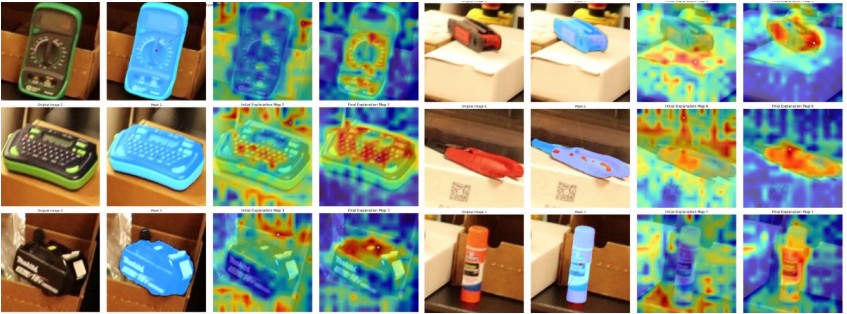

Figure B1: Explainability maps for optimized token. Columns from left to right. (1) The cropped object. (2) ground-truth mask (3) Explainability map before optimization in MaO-CLIP (4) Explainability map after our multi-object optimization with MaO-CLIP. The explainability attend to the objects after optimization (warmer colors coincide the object areas). A single optimized token can capture multiple objects across different input object crops.

## C Evaluation on non-small Objects and more Datasets

To further validate the generalization of our approach, we conducted additional experiments on a new dataset, ILIAS [13], which includes a mix of small, medium, and large objects. We further add MaO also with an additional backbone of SIGLIP. Note that ILIAS also includes small objects among the gallery images. The ILIAS dataset consists of 1,000 unique classes, 1,232 query images, and 4,715 positive gallery images. It is available in two versions: one with 5 million distractor images and another with 100 million distractors. We report results and comparisons on the version without distractors avoiding heavy computation yet showing clear differences between various approaches. In addition to the baselines, we also include results for Stage A, as defined in the main paper, in the zero-shot setting and also after applying multi-object fine-tuning in our supervised setting.

Table C1 compares our method to the strongest baseline, GeM, as well as to DINOv2 and SIGLIP models whose results are reported in [13]. Specifically, we include DINOv2-Base and SIGLIP-Base (without linear adaptation), and DINOv2-Large[†] and SIGLIP-B[†] (with linear adaptation). Note that the reported values for all baselines from [13] are mAP@1K, which tends to be higher than the true mAP computed over the entire gallery. We follow the same performance measure to allow comparison.

Although the extensive experiments in the main paper include tests on common backbones and SoTA instance based retrieval methods, we additionally present results of MaO using the also the SIGLIP backbone. We observe that incorporating our multi-object optimization (MaO) into SIGLIP leads to a substantial performance gain also on this strong backbone. The results indicate that our method significantly outperforms all baselines in both zero-shot and fine-tuned settings with the DINOv2 backbone. Furthermore, applying MaO to SIGLIP in Zero-Shot manner achieves top performance even when compared to the fine-tuned linear adaptation model.

**MaO Performance on large and general image content–based retrieval:** We conducted an additional evaluation on the PODS dataset [41], which has an average object-to-image area ratio of 20%. As shown in Tab. C2, MaO performs robustly on these larger instances, with no degradation in performance.

## D Comparison to MLLM

Although current multimodal large language models (MLLMs) such as Qwen or Ferret are not designed for large-scale image retrieval, we still provide an evaluation for comparison. We can operate such models by requiring pairwise checks between the query image and each gallery image, e.g., prompting whether the object appears in a given image, with both images passed through the full MLLM pipeline. This approach is computationally costly and unsuitable for large-scale scenarios, yet as an interesting experiment we illustrate the performance of Qwen2.5-VL on 100 queries from a VoxDet subset considering a gallery of 200 images.

Table C1: Retrieval performance (mAP) on the ILIAS dataset (without distractors). Results are shown for both zero-shot and fine-tuned settings. Stage A refers to our average pooling of object embeddings in the zero-shot setting and multi-object fine-tuning in the supervised setting. [†] Indicates results with linear adaptation. B,L indicate Base and Large backbone versions respectively.

|  | Method | mAP |
|---|---|---|
| Zero Shot | GeM | 30.20 |
|  | SIGLIP-B | 20.20 |
|  | DINOv2-B | 14.30 |
|  | DINOv2-B (Stage A) | 36.41 |
|  | MaO-DINOv2-B | 38.93 |
|  | MaO-SIGLIP-B | **62.22** |
| Fine Tuned | GeM | 34.46 |
|  | SIGLIP-B[†] | 58.70 |
|  | DINOv2-L[†] | 46.50 |
|  | DINOv2-B (Stage A) | 45.29 |
|  | MaO-DINOv2-B | 48.85 |

Table C2: Results (with finetuned models) on non-small objects, PODS dataset.

| Method | mAP |
|---|---|
| CLIP | 0.341 |
| DINOv2 | 0.332 |
| MaO-CLIP | 0.368 |
| MaO-DINOv2 | 0.393 |

Table D1: Comparison of Qwen2.5-VL and MaO-DINOv2 on a subset of VoxDet.

| Method | Matching Type | Accuracy |
|---|---|---|
| Qwen2.5-VL | Pair Matching | 19% |
| MaO-DINOv2 | Single Vector Retrieval | 85% |

These results show that, beyond its impractical runtime, an MLLM like Qwen struggles with small-object retrieval, highlighting the advantages of dedicated retrieval models such as MaO.

# E  Additional Ablation Study

## E.1  Multi-Object Optimization

We present additional ablation studies to examine the effect of the regularization term on performance. Figure E1 presents the mAP results for MaO-CLIP and MaO-DINOv2 on VoxDet$_\mathcal{W}$ dataset across different values of $\alpha$ in Eq. (1) of the main paper. We set the optimal value as 0.03 across all of our benchmarks, according to Figure E1.

We further examine the impact of the number of iterations in our refinement stage (Stage B). Figure E2 presents the result as a function of the iteration count. We therefore fixed the number of iterations to 80 in all of our benchmarks. This stage typically takes 0.03 sec per-object, per-image.

# F  Limitation Analysis

**Dependence on OVD:** Our proposed approach relies on the performance of the OVD. In this section we evaluate the used OVD of OWLv2 [22] on the final performance in different aspects. As our model uses an OVD in its first stage it has an inherent limitation of any modular, detection-first retrieval pipeline. Two-stage designs are common in computer vision and are well-established paradigm that is widely adopted in various methods as seen e.g. in SAM [12] that uses an OVD as its first stage, *Grounded-SAM* [33] (combining Grounding DINO OVD), *Open3DIS* [23] that performs

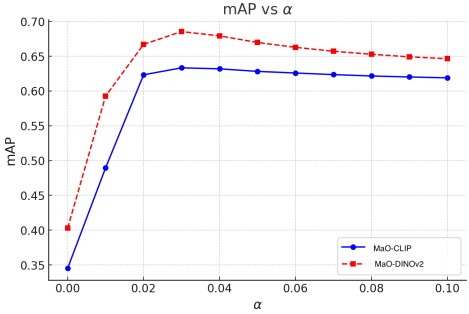

Figure E1: The mAP performance of MaO-CLIP and MaO-DINOv2 as a function of the regularization weight $\alpha$ examined on VoxDet$_\mathcal{W}$

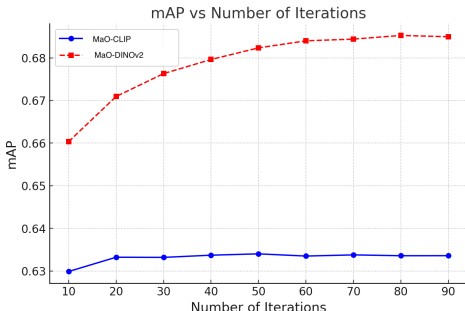

Figure E2: The mAP results as a function of the iteration count in our multi-object optimization stage for the VoxDet$_\mathcal{W}$ dataset.

open-vocabulary 3D instance segmentation, and *TROY-VIS* [49] that uses an existing OVD for video instance tracking, or in many object detectors that rely on object proposal methods. In our case, the OVD is used purely as an objectness (object proposal) stage. We further presented an analysis showing that modern OVDs (e.g., OWLv2) are capable of detecting small and even unusual objects effectively.

Next, in Table F1 we evaluate the Recall values in our setting. We present the results with different predicted bounding-box vs. tight ground truth overlaps on VoxDet, measured by $IoU$ threshold. Results are provided for two overlap thresholds $IoU = 0.6, 0.9$. We observe that relaxing the overlap constraint for a True Positive significantly increases Recall, indicating that while OVD performs well in object detection, it may lack precision in tightly localizing objects. However, since our approach applies center-cropping around each predicted object, often with a margin, particularly for small objects (avoiding resize), it effectively captures the object even when localization is imprecise. The subsequent mask optimization process further refines the representation, ensuring a more precise encoding even if the object is not perfectly centered within the crop. The average number of detected objects per scene further reflects the complexity of the benchmarks. While INSTRE contains fewer than five detected objects per image, VoxDet and PerMiR feature more than ten, highlighting their higher level of clutter.

Table F1: Our OVD (OWLv2) detection performance. We report the average number of objects detected per image. For True-Posiitve detection is defined as $IoU \geq 0.9$.

In case of $IoU = 0.6$ we further relax the condition on objects contained in the predicted bbox by setting a containment ratio threshold of $0.3$ for true positive.

|  | Avg. #Objects | Recall[%] |
|---|---|---|
| **VoxDet$_\mathcal{W}$**$(IoU = 0.9)$ | 14.7 | 63 |
| **VoxDet$_\mathcal{W}$**$(IoU = 0.6)$ | 14.7 | 96 |
| **INSTRE-XS** | 3.8 | 87 |
| **INSTRE-XXS** | 4.2 | 83 |
| **PerMiR** | 10.43 | 88 |

Novel OVD methods can operate even under extreme object size and clutter conditions (see Table F2), as we leverage this in our approach.

**The impact of the OVD on MaO is limited and measurable:** To further address this concern, we conducted an empirical evaluation of our chosen OVD model on two GT-annotated benchmarks and reported the results in Tab. F2. The results show that even under imperfect detection (partial object detection) our method continues to outperform existing approaches.

Table F2: OVD performance across varying object-to-image area ratios, on VoxDet-SoR.

| Object-to-image area Ratio (%) | 0.2 | 0.4 | 0.8 | 1.0 | 1.5 | 2.0 |
|---|---|---|---|---|---|---|
| Object Recall | 0.68 | 0.83 | 0.83 | 0.85 | 0.90 | 0.87 |

While there is a performance degradation in the extreme case of 0.2%, the results show the high capability of `owlv2` in discriminating even small objects.

**Performance of the OVD in cluttered scenes**

In Tab. F3 we provide a breakdown of the used OVD (on VoxDet-SoR) according to several clutter levels. The OVD maintains strong performance even in extreme cases where 12–24 objects are detected in the image. We further analyze the impact of clutter level on retrieval performance. To systematically evaluate performance under increasing clutter, we conduct an experiment where the number of objects in the gallery images is gradually increased. Using the GT masks, we encode positive images by combining the representation of the target object with those of randomly selected negative objects. At each step, as more objects are added, we incrementally incorporate additional object embeddings into our MaO when forming the global image representation. The results of this experiment are presented in Figure F1 and confirm a moderate performance drop with gradually increased clutter (up to 6 objects), decreasing from 0.96 mAP for a single object to approximately 0.82 mAP when six objects are present. This further highlights MaO capability in handling multi-object representation.

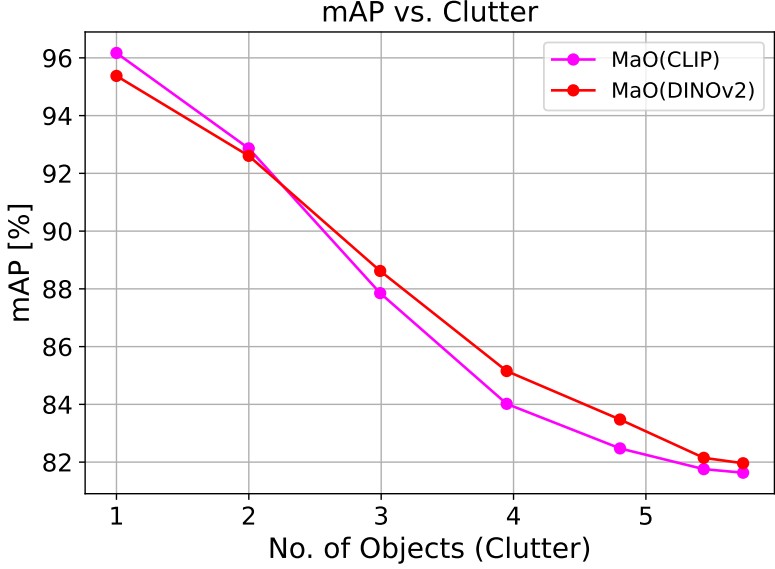

Figure F1: Impact of scene clutter on retrieval. As expected Increased scene clutter negatively affects accuracy.

Table F3: Recall of the OVD across different numbers of detected objects (on VoxDet-SoIR).

| # Objects | 12 | 16 | 20 | 24 |
|---|---|---|---|---|
| Recall | 0.80 | 0.95 | 0.95 | 0.89 |

In summary, we believe our method effectively harnesses the strengths an OVD provides for image retrieval, while addressing and mitigating its inherent limitations.

**Partial Detection, Occlusion and Heavy Clutter:** An example of partial detection is shown in Figure F2, where the OVD detects only part of the target image. Scenes with high clutter and

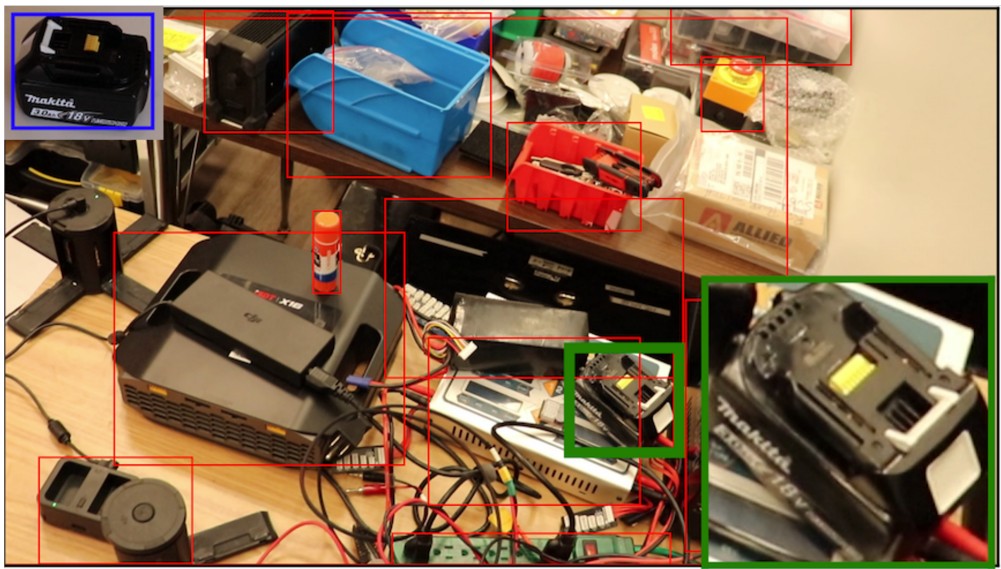

Figure F2: Retrieval failure likely due to partial object detection. The green box is the ground truth, while red boxes are OVD detections, none covering the full object. The object occupies 1% of the image area.

numerous objects present additional challenges to our model. In cases where the OVD detects more than 25 items, we often find that the target object, although detected, may be under-represented and thus missed during retrieval. Occlusion and overlapping objects further increase the sensitivity for small objects, making it difficult to extract distinctive features necessary for accurate retrieval.

Finally, we further noticed that our model is more biased toward objects and has shortcomings on faces, individuals, and landmarks.

## G    More Qualitative Results

Additional qualitative comparisons on tiny objects in INSTRE-XXS are presented in Figures G1 and G2. We provide examples comparing our approach with strong fine-tuned versions of DINOv2 and CLIP, as well as the state-of-the-art instance retrieval method of SuperGlobal [37]. While DINOv2 and CLIP each retrieve only one correct image in Figure G1 in all other results in Figures G1 and G2 they fail to retrieve the correct images within the top-5 ranks. In contrast, our approach consistently retrieves the correct images from the gallery.

In Figure G3 we further present comparative results on VoxDet$_\mathcal{W}$.

## H    Image Sampled from PerMiR Dataset

Figure H1 presents examples of multiple samples from the PerMiR dataset, the query (at top) and associated gallery images (at the bottom). Since PerMiR was constructed from videos, some cases can be easy and some extremely hard.

## I    Additional Analysis of INSTRE

To further highlight the differences between INSTRE-XS, INSTRE-XXS, and the standard INSTRE-S1 and INSTRE-M benchmarks, we display the largest objects from the INSTRE-S1, INSTRE-M, INSTRE-XS, and INSTRE-XXS subsets in Figure I1, demosntrating the significant difference in the object size, between the benchmarks. Additionally, we present the distribution of object sizes for each benchmark in Figure I2, illustrating the importance of our subset selection for small object tasks. Note that INSTRE-XS and INSTRE-XXS impose upper size limits of 15% and 5%, respectively.

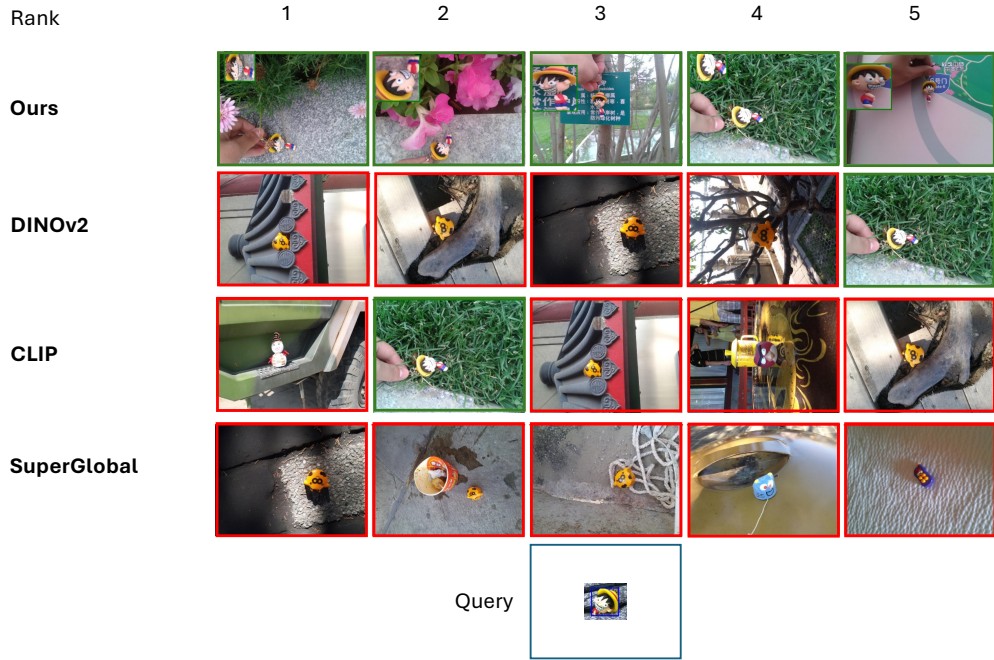

Figure G1: Qualitative results on INSTRE-XS and INSTRE-XXS, comparing retrieval performance across MaO vs. fine-tuned DINOv2, CLIP, and state-of-the-art SuperGlobal. Green and red borders indicate correct and incorrect results respectively. The query input image is shown at the bottom. Zoomed object images are overlaid retrieval image for better visibility. Except for one correct image retrieved by DINOv2 and one by CLIP, all methods fail to retrieve the correct small target images within the top-5 ranks. In contrast, MaO method successfully retrieves the true target images in all top-5 results. The relative object sizes for the correct top-5 gallery images retrieved by our method are 1.24%, 1.93%, 1.21%, 2.1%, and 1.01%, respectively.

## J  Additional Analysis of VoxDet

We provide additional insights into the train and test datasets of VoxDet-SoIR. The figures illustrate samples of five instances, where the query images are displayed at the top, and their corresponding gallery images are shown at the bottom. Figure J1 presents examples from the training set, while Figure J2 showcases samples from the test set. While the training dataset includes small objects within complex scenes containing multiple objects and significant occlusions, the test dataset features even smaller objects in realistic, cluttered environments with fewer occlusions. Figures J3 and J4 illustrate the distribution of the number of objects per gallery image in the training and test datasets, respectively. Similarly, Figure J5 depicts the relative sizes of objects within each gallery image for the training and test datasets, respectively. Although the test dataset exhibits less variability in the number of objects per image, the objects themselves appear even smaller compared to those in the training dataset.

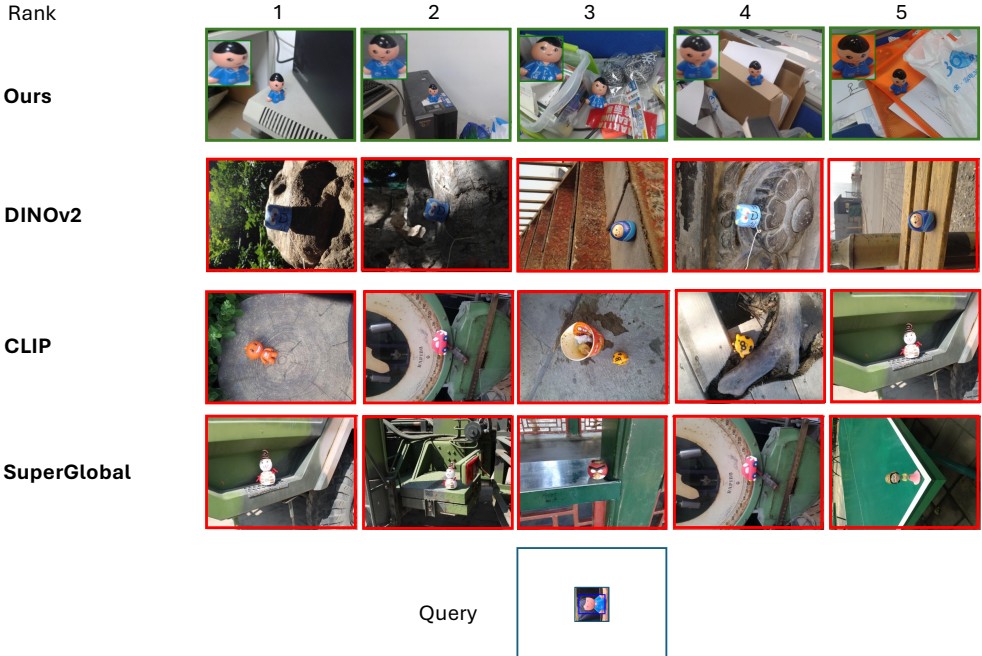

Figure G2: Qualitative results on INSTRE-XS and INSTRE-XXS, comparing retrieval performance across MaO vs. fine-tuned DINOv2, CLIP, and state-of-the-art SuperGlobal. Green and red borders indicate correct and incorrect results respectively. The query input image is shown at the bottom. Zoomed object images are overlaid retrieval image for better visibility. All other methods fail to retrieve the correct small target images within the top-5 ranks. In contrast, MaO successfully retrieves the true target images in all top-5 results. The relative object sizes for the correct top-5 gallery images retrieved by our method are 2.36%, 1.03%, 2.59%, 1.82%, and 2.08%, respectively.

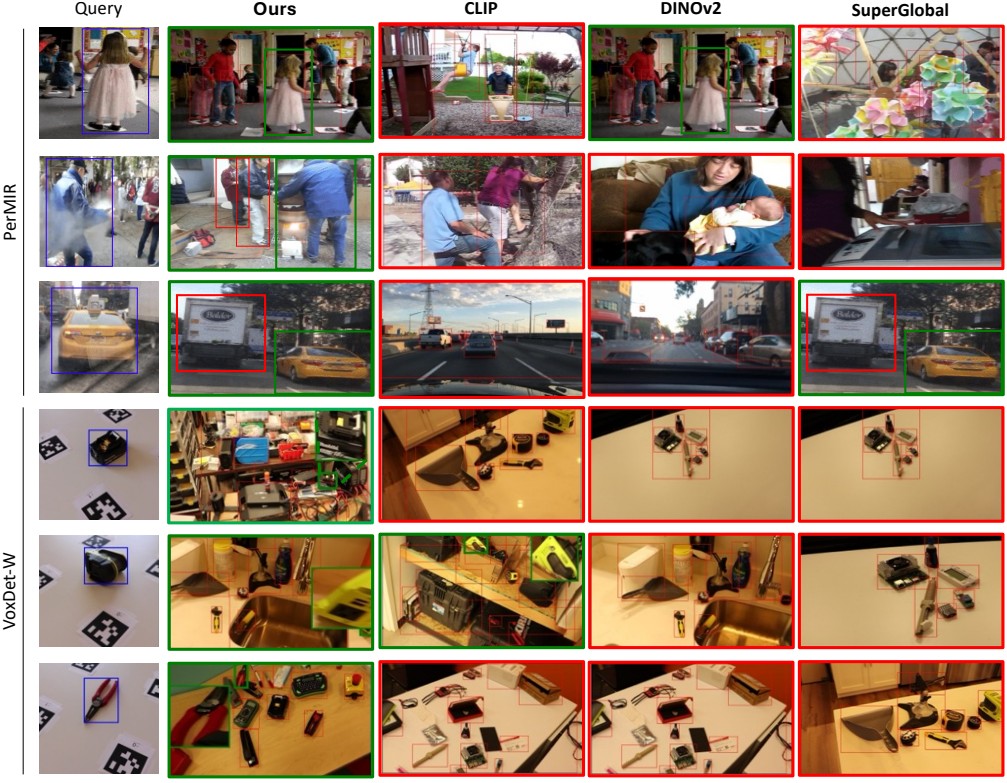

Figure G3: Results using various methods on VoxDet$_\mathcal{W}$ and PerMiR$_\mathcal{W}$, showing Rank-1 results. Red and green frames indicate incorrect and correct retrievals, respectively. Red boxes in the image denote OVD detections. Notice the high level of clutter caused by frequent OVD detections in the gallery images. Zoom-in objects are depicted at the edge to enhance visibility. The target object size for the first three lines in PerMiR$_\mathcal{W}$ are 20%, 42%, and 32% of the image area, respectively, while in VoxDet$_\mathcal{W}$, target objects cover 0.85%, 0.17%, and 0.51% of the image area.

## Sample images from PerMiR

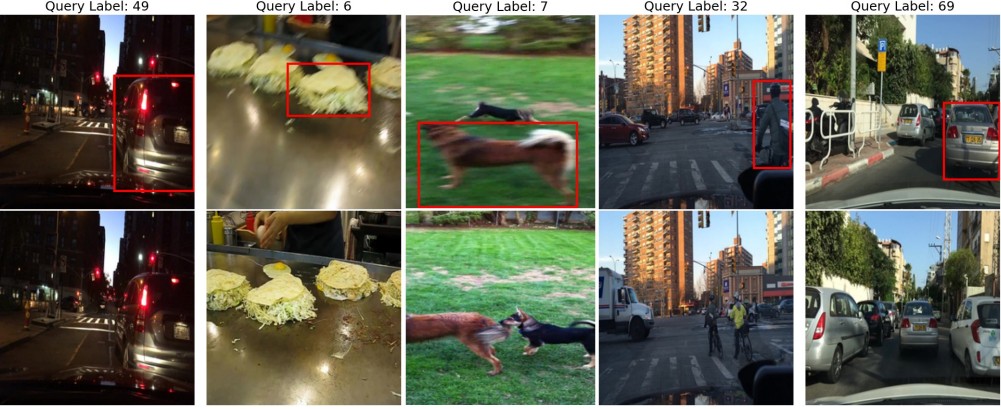

Figure H1: Example samples from the PerMiR dataset. The top row in each subfigure represents the query images, while the bottom row shows the corresponding gallery images containing the retrieved instances. The red bounding boxes indicate the objects in the query images. Gallery images typically contain multiple objects and often from the same catergory, several cars, food items or dogs.

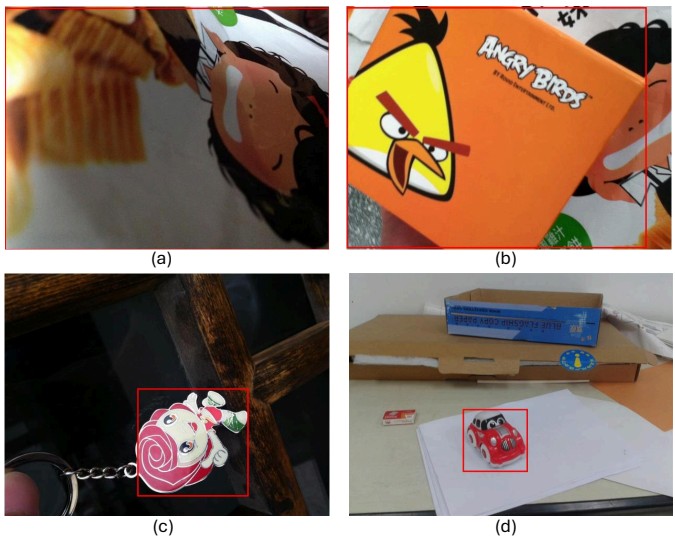

Figure I1: Largest Objects in the INSTRE Datasets: Examples of the largest objects are presented from the following subsets: (a) INSTRE-S1, (b) INSTRE-S2, (c) INSTRE-XS, and (d) INSTRE-XXS, with their respective relative sizes being 99.26%, 90.7%, 15%, 4.99%

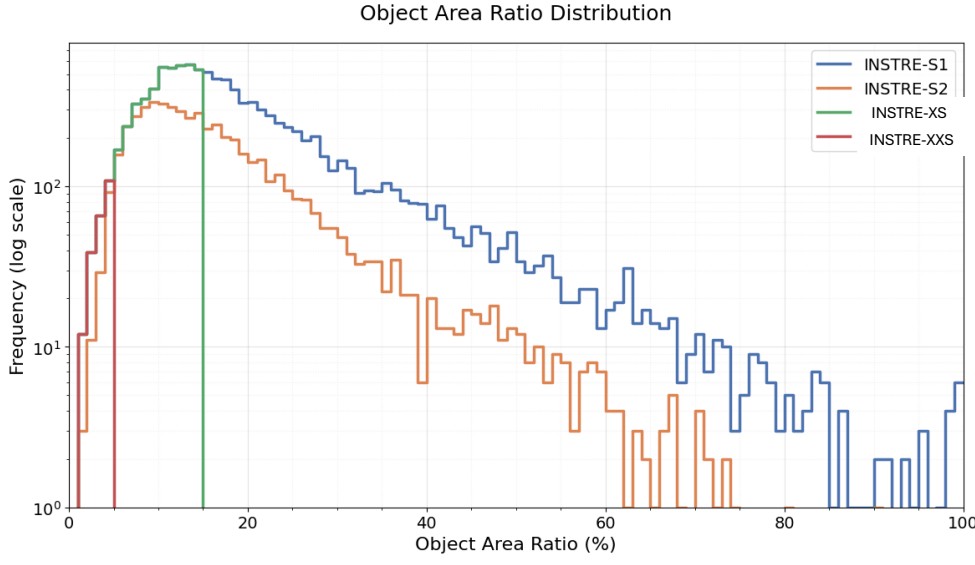

Figure I2: Distribution of Object-Image area ratio across the four INSTRE subsets: INSTRE-S1, INSTRE-S2 and our selected subsets INSTRE-L and INSTRE-T. Notably, INSTRE-XS and INSTRE-XXS are limited to small object size, allowing a better performance assessment on this specific task.

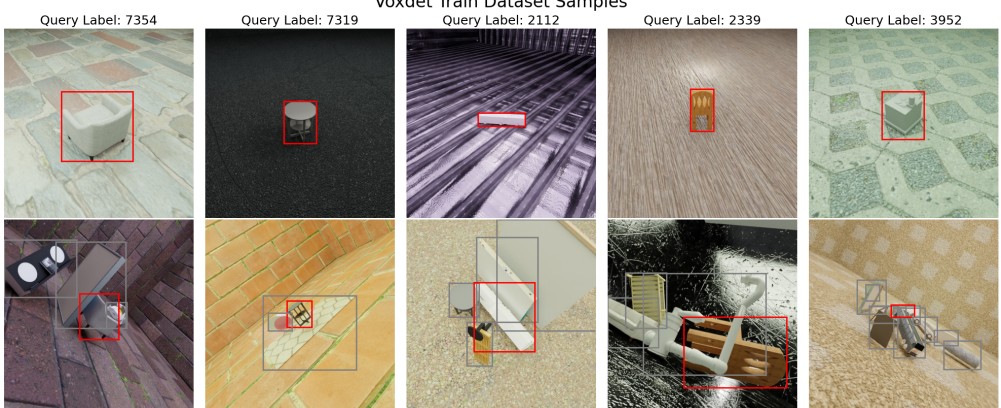

Figure J1: Example samples from the VoxDet training dataset. The top row displays query images, and the bottom row presents gallery images containing the corresponding instances.

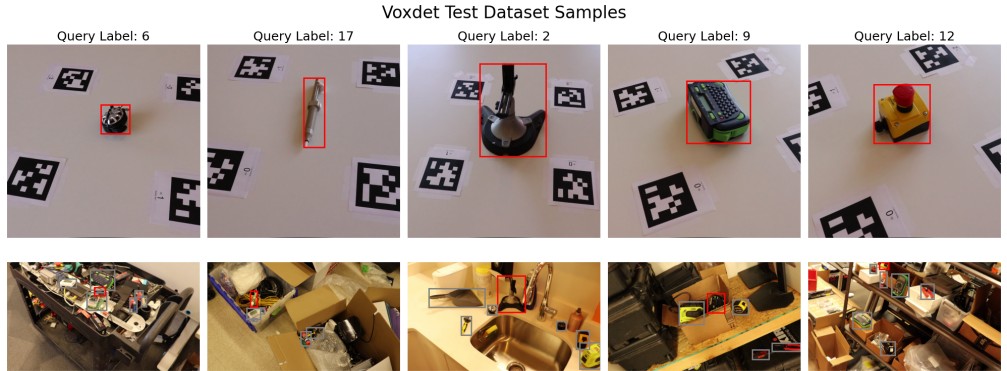

Figure J2: Example samples from the VoxDet test dataset. Similar to the training set, the top row contains query images, while the bottom row includes gallery images with matching instances.

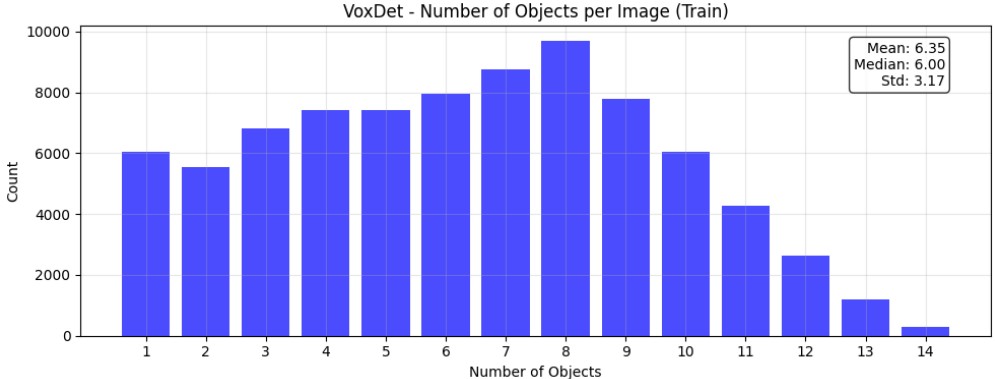

Figure J3: Distribution of the number of objects per gallery image in the VoxDet training dataset. The dataset exhibits a range of object counts per image, with a mean of 6.35 objects, a median of 6, and a standard deviation of 3.17.

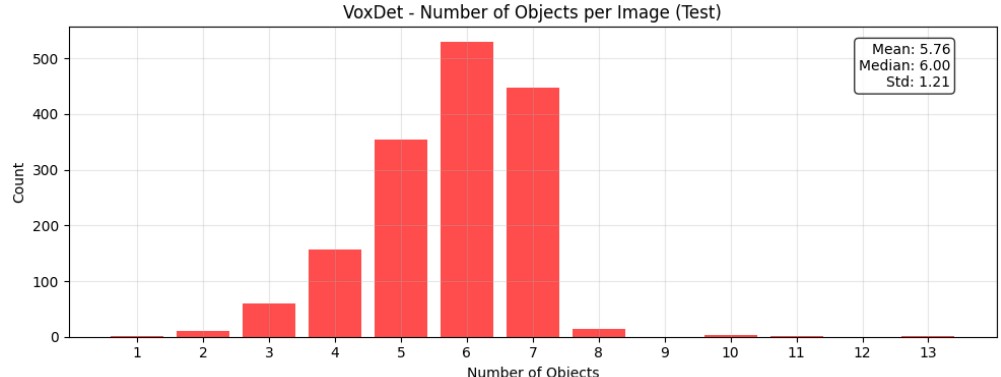

Figure J4: Distribution of the number of objects per gallery image in the VoxDet test dataset. The number of objects varies similarly to the training set but may reflect differences in scene complexity.

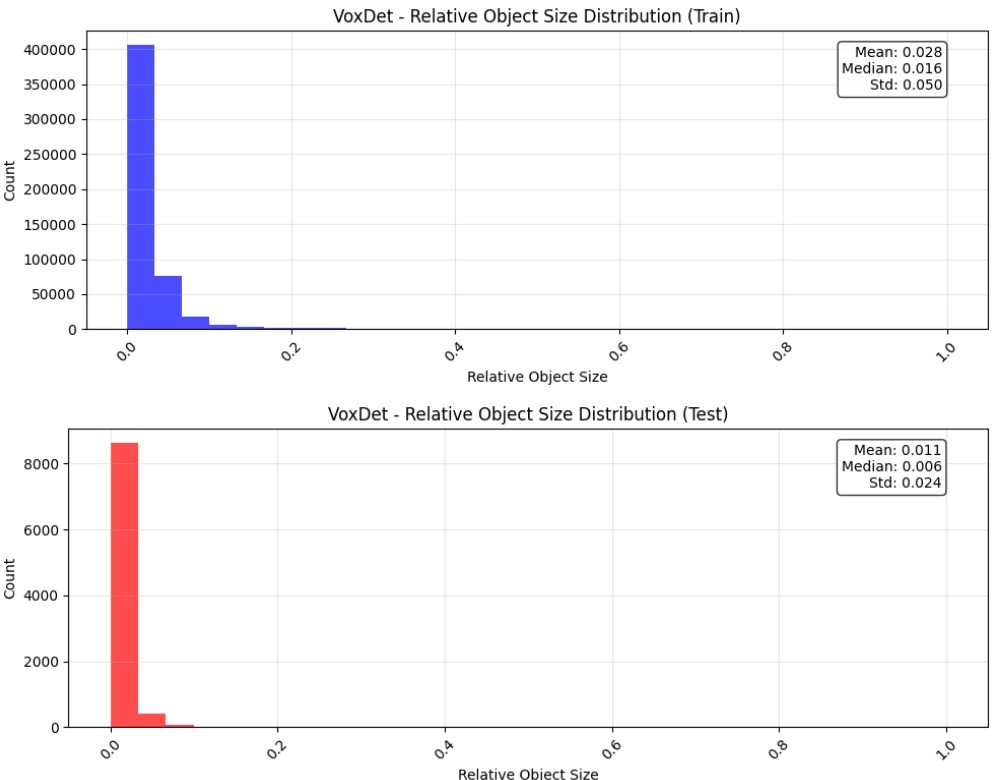

Figure J5: Top: Relative object sizes - Object to image area ratio, within each gallery image in the VoxDet training dataset. The figure illustrates the proportion of the object size relative to the image dimensions. Bottom: relative object sizes within each gallery image in the VoxDet test dataset.

