# OpenReview forum: "Find your Needle: Small Object Image Retrieval via Multi-Object Attention Optimization"
_NeurIPS.cc/2025/Conference — NeurIPS 2025 poster_

### Official Review · Reviewer_1h7Q · 2025-07-03

**Clarity:** 2
**Significance:** 2
**Originality:** 2
**Rating:** 3
**Confidence:** 4

**Summary:**

This paper tackles the problem of Small Object Image Retrieval (SoIR), which involves retrieving images containing specific small objects within cluttered scenes. Existing methods struggle to represent multiple small objects effectively using a single image descriptor. To address this, the authors propose Multi-object Attention Optimization (MaO)—a new retrieval framework that includes multi-object pre-training and attention-based feature refinement using object masks. MaO produces a unified image descriptor that improves retrieval accuracy, especially in zero-shot and lightweight fine-tuning settings.

**Questions:**

Please refer to the weakness section.

**Ethical Concerns:**

["NO or VERY MINOR ethics concerns only"]

**Final Justification:**

The rebuttal addressed some of my concerns with clarification and additional ablation, such as other attention aggregation schemes. However, I am still not convinced by the incremental novelty. I would like to keep my rating.

**Limitations:**

These items are discussed in the paper.

**Paper Formatting Concerns:**

I did not notice any issues.

**Quality:**

2

**Strengths And Weaknesses:**

Strengths

•	One contribution of this paper is the introduction of new benchmarks (subsets of INSTRE and a retrieval version of VoxDet) that specifically target the weaknesses of current methods concerning small and cluttered objects.

•	The paper presents experiments that demonstrate the superiority of MaO over a wide range of baselines, including both classic image retrieval methods and modern foundation models

Weaknesses

•	The MaO pipeline is critically dependent on the performance of an upstream open-vocabulary object detector (OVD) to first identify objects in the gallery images. The paper acknowledges that the method's performance can be affected by imperfect detector recall. This is a significant limitation, as the method would likely fail if the target object is missed by the detector. A more thorough analysis of this failure mode and its impact on performance would strengthen the paper.

•	The Multi-Object Attention Optimization (Stage B) is an iterative process that must be performed for every image in the gallery to generate its final descriptor. The paper notes this is an offline process, but at 0.03 seconds per object and with 15 objects per image on average for VoxDet, the overhead is substantial (0.5 seconds per image). This is significantly more expensive than the single forward pass required by most baselines and may pose scalability challenges for very large galleries.

•	Overall, the novelty of the proposed method appears to be incremental. The approach of using an object detector to first isolate regions of interest is an established pattern in image retrieval. More significantly, the core technical mechanism in Stage B, which involves refining a representation by optimizing its attention-based explainability map against an object mask, is heavily inspired by recent work like LeGrad and MaskInversion (regularization term), as the authors acknowledge.

•	During the multi-object fine-tuning (Stage A), the representation for a gallery image is created by simple average pooling of all its detected object features. In a cluttered scene with many distractor objects, this naive averaging could dilute the features of the actual target instance. The paper would benefit from a justification for this choice or an ablation study comparing it to other aggregation mechanisms (e.g., attention-based pooling).

•	The paper's core premise is the superiority of its single-vector representation for scalability. To better contextualize this, it would be highly informative to include a comparison against a multi-vector baseline (e.g., where each object crop is a separate vector in the index). This would help quantify the accuracy-efficiency trade-off and demonstrate how close MaO's optimized single vector comes to the performance of a less efficient but potentially more powerful representation.

---

> ### Author Rebuttal · Authors · 2025-07-30
>
> Thank you for your valuable review and recognizing **“the superiority of MaO”**, and the contribution of **“new benchmarks”**.
>
> ## General Concerns:
> ### **“The MaO pipeline is critically dependent on an upstream OVD.**
>
> This raised concern is an inherent limitation of any modular, detection-first retrieval pipeline, and we appreciate the opportunity to address it in more depth.
>
>
> 1. Dependence on an OVD is a standard practice across several domains.
> Our design follows a well-established paradigm that is widely adopted in various methods e.g., Grounded-SAM (combining Grounding DINO OVD), Open3DIS (CVPR 2024) that performs open-vocabulary 3D instance segmentation, and TROY-VIS (WACV 2025) that uses an existing OVD for video instance tracking.
> Thus, the use of an OVD as a building block is not unique to our method, but consistent with many best practices in the field. Novel OVD methods can operate even under extreme object size and clutter conditions (see table below), and we leverage this in our approach.
>
>
> 2. No need for object classification, just object proposals
> We use OVD in the “objectness” mode (L207-208) where the model is used as an object proposal and needs to binary discriminate an image area as object or background.
>
> 3. In practice, the impact is limited and measurable.
> To further address this concern (that was discussed in Section 5), we conducted an empirical evaluation of our used OVD model on two GT-annotated benchmarks, and reported in our  Suppl. Material (SM) in Section F (as referred in L350 in the main paper). The results show that even under imperfect detection (partial object detection) our method continues to outperform existing approaches.
> In summary, we believe our method effectively harnesses the strengths an OVD suggests for image retrieval, while addressing and mitigating its inherent limitations.
>
> | Object-to-image area ratio (%) | 0.2 | 0.4 | 0.8 | 1.0 | 1.5 | 2.0 |
> |-|-|-|-|-|-|-|
> | Object Recall               	| 0.68 | 0.83 | 0.83 | 0.85 | 0.90 | 0.87 |
>
> While there is a performance degradation in the extreme case of 0.2% the results show the high capability of owlv2 in discriminating even small objects.
>
> ### **Discussion on failure cases**
> Thank you. Beyond our discussion on limitations in Section 5 and Sections F and H in the Suppl. Material (see pointer to this section in L335-336, including OVD and clutter), we further noticed that our model is more biased toward objects and has shortcomings on faces, individuals, and landmarks. We will extend and add visual examples (omitted here due to formatting constraints) in the final version.
>
> ## Individual Concerns:
>
> ### **[W2]  Runtime of MaO iterative process**
> Thank you for this comment. Most state-of-the-art retrieval methods adopt either a two-stage design with re-ranking (e.g., SuperGlobal) or rely on computationally intensive local matching (e.g., AMES), making them significantly less efficient during online search.
> We present a detailed runtime analysis in Section D of the Supplementary Material, comparing our method against SuperGlobal (SG) and AMES.
> For offline encoding, both SG and AMES require 65 ms per image, while MaO requires 174 ms per image, plus an additional 0.5 seconds per image for the OVD step. However, this OVD overhead can be substantially reduced through batch processing or optimized implementations such as ONNX. It’s worth noting that the entire MaO encoding pipeline is highly data-parallelizable, offering further potential for acceleration.
> The online test-time advantage of MaO is clear: it requires only 30.03 ms per query, compared to 80.4 ms for SG and 2,286 ms for AMES (evaluated on VoxDet with 1,581 gallery images). While there is an encoding-time trade-off, we believe it can be effectively addressed through code-level optimization and parallelization.
>
> ### **[W3] The approach of using an object detector to first isolate regions of interest …**
> While the use of object detectors to isolate regions of interest is not new, our contribution lies in how this component is integrated into our new complete retrieval framework, effectively demonstrating its strength in retrieving small objects within cluttered scenes.
> Our contribution stems from the insight that, for retrieving small objects in cluttered scenes, isolating individual objects significantly improves their representation, leading to more balanced and effective aggregation.
>
> ### **[W3] … the core technical mechanism in Stage B …**
> Our new aggregation model leveraging Mask-Inversion in fact allows handling multiple objects in the image.  This addresses a critical gap in prior Mask Inversion methods, which neither target retrieval nor support multiple-object encoding. Our experiments demonstrate that this component is especially beneficial for small object retrieval, an area often under-explored by existing techniques. In our solution, we optimize a single vector against K object-level representations, backpropagating gradients only through the aggregated vector (see Fig.2).
> An additional novelty, we believe, is the application of Stage-B refinement on a customized multi-object encoder trained in Stage-A, making the two stages tightly integrated and complementary.
>
> As detailed in our contributions, our work presents several distinct and non-trivial components across methodology, benchmarking, and analysis.
>
> ### **[W4] Comparison to Other Aggregation Mechanisms (e.g., Attention-based Pooling)**
> Thank you for the valuable suggestion. While our use of average pooling is intentionally simple and non-parametric, to enable zero-shot applicability (as shown in the paper), it ensures broad and simple compatibility with different backbones, and highlights the strength of our MaO approach even with minimal aggregation complexity.
> That said, we agree that more advanced aggregation methods may further enhance performance. In response to the reviewer’s suggestion, we evaluated our Stage-A model using a single-layer attention-based pooling mechanism. The results show a performance improvement, indicating that our framework is flexible and can effectively incorporate alternative aggregation strategies.
>
> **map on VoxDet:**
> | Aggregation Method      | mAP (%) |
> |-|-|
> | Average Pooling         | 75.04   |
> | Attention-based Pooling | 78.17   |
>
>
> We appreciate the reviewer’s input. Exploring more advanced pooling techniques is a promising direction for future work.
>
> ### **[W5] Multivector baseline ….**
> Thank you. This multi-vector baseline demonstrates a slight performance gain over our MaO single-vector representation, but comes at a substantial cost. For VoxDet, a nearly 6× increase in gallery size and memory usage during search (please see discussion in L22-24). While this strategy may be useful as a potential re-ranking stage in future work, it would introduce additional search latency.
>
> Results on VoxDet with DINO (Fine-Tuned)
> | Method                	| Avg. Vectors/Image | mAP (%) |
> |-|-|-|
> | DINO (Multi-Vector)   	| 5.76           	| 85.40   |
> | MaO-DINO (Single-Vector)  | 1.00           	| 83.70   |

---

> > ### Comment · Reviewer_1h7Q · 2025-08-05
> > **post rebuttal**
> >
> > Thanks to the authors for the rebuttal. The rebuttal addressed some of my concerns with clarification and additional ablation, such as other attention aggregation schemes. However, I am still not convinced by the incremental novelty. I would like to keep my rating.

---

> ### Author Response · Authors · 2025-08-05
>
> Thank you for your time and effort in reviewing our paper and the constructive remarks.
>
>
> Regarding the remaining concern, we respectfully refer to the [NeurIPS 2025 Reviewer Guidelines](https://neurips.cc/Conferences/2025/ReviewerGuidelines), where the Originality criterion states:
>
> > "Does the work introduce novel tasks or methods that advance the field? Does this work offer a novel combination of existing techniques, and is the reasoning behind this combination well-articulated?  As the questions above indicates, originality does not necessarily require introducing an entirely new method. Rather, a work that provides novel insights by evaluating existing methods, or demonstrates improved efficiency, fairness, etc. is also equally valuable.  As the questions above indicates, originality does not necessarily require introducing an entirely new method. Rather, a work that provides novel insights by evaluating existing methods, or demonstrates improved efficiency, fairness, etc. is also equally valuable."
>
> We believe our work goes beyond what the guidelines describe.

---

### Official Review · Reviewer_Ey8L · 2025-07-07

**Clarity:** 2
**Significance:** 2
**Originality:** 2
**Rating:** 4
**Confidence:** 5

**Summary:**

This paper presents a two-stage framework for small object image retrieval. The motivation is to retrieve images containing a specific, often tiny, object within cluttered scenes using a single global descriptor. In first stage, it learns a representation for multiple objects in the image by cropping and encoding objects separately and aggregating them via average pooling to align with object queries. The second stage is a post-training phase where learnt representations are enhanced via multi-object attention mechanism. Author introduces new benchmarks focused on small-object retrievals  by deriving new datasets (Instre-XS). Experiments on the  variety of datasets show improved performance over the counterpart methods

**Questions:**

I have listed all my concerns in the "Strength and Weaknesses section". Please refer to "Weaknesses" for the raised concerns and questions.

**Ethical Concerns:**

["NO or VERY MINOR ethics concerns only"]

**Final Justification:**

I would like to thank the authors for their responses. I have read reviews and responses. I have increased my rating to borderline accept. I also agree with the other reviewers on the amount of novelty part. Additional novel contributions on reducing the reliance on OVD and improving computational efficiency would make this paper stronger.

**Quality:**

2

**Strengths And Weaknesses:**

Strengths

1)	The paper focuses on a rather unexplored area in the image retrieval domain, retrieval of very small objects (down to 0.5% of image area) in cluttered  scenes.

2)	Two-stage framework , multi-object fine-tuning and multi-object attention mechanism, provides a systematic approach to address the problem.

3)	The release of new benchmarks dedicated for small object retrieval, which can be used for future studies.

4)	Extensive experiments showing the improved performance over the counterpart methods

Weaknesses

1)	The performance heavily depends on the  open vocabulary detector. It is not clear how the model would handle rare or out of domain objects.

2)	In stage 1, it is unclear how objects are retrieved. How do you prompt the model, and how do you ensure small and rare objects, i.e. the object of interests, are correctly cropped and encoded.

3)	In stage 1, it seems all type of objects are cropped and encoded. The small objects or rare objects are generally underrepresented. How can proposed approach avoid the bias towards frequent/large objects?

4) In second stage, why not fine-tune vision encoder to learn better representations for smaller objects?

5)	How does the proposed method compares with more advanced approaches such as Qwen (or Ferret) models that are capable of referring expression and grounding, and expected to generalize over rare/small objects.

6)	No information on computational complexity of proposed method compared to counterpart methods.

7)	What are the failure cases. A deeper analysis of failure cases would enable the readers to understand the limitations.

---

> ### Author Rebuttal · Authors · 2025-07-30
>
> Thank you for your valuable review and for recognizing that our work demonstrates **“improved performance”**, and **“provides a systematic approach”** with **“extensive experiments”** addressing an **“unexplored area”**, as well as acknowledging the contribution of a “benchmark that can be used for future studies.”
>
> ## General Concerns:
> ### **“The MaO pipeline is critically dependent on an upstream OVD.**
>
> This raised concern is an inherent limitation of any modular, detection-first retrieval pipeline, and we appreciate the opportunity to address it in more depth.
>
>
> 1. Dependence on an OVD is a standard practice across several domains.
> Our design follows a well-established paradigm that is widely adopted in various methods e.g., Grounded-SAM (combining Grounding DINO OVD), Open3DIS (CVPR 2024) that performs open-vocabulary 3D instance segmentation, and TROY-VIS (WACV 2025) that uses an existing OVD for video instance tracking.
> Thus, the use of an OVD as a building block is not unique to our method, but consistent with many best practices in the field. Novel OVD methods can operate even under extreme object size and clutter conditions (see table below), and we leverage this in our approach.
>
>
> 2. No need for object classification, just object proposals
> We use OVD in the “objectness” mode (L207-208) where the model is used as an object proposal and needs to binary discriminate an image area as object or background.
>
> 3. In practice, the impact is limited and measurable.
> To further address this concern (that was discussed in Section 5), we conducted an empirical evaluation of our used OVD model on two GT-annotated benchmarks, and reported in our  Suppl. Material (SM) in Section F (as referred in L350 in the main paper). The results show that even under imperfect detection (partial object detection) our method continues to outperform existing approaches.
> In summary, we believe our method effectively harnesses the strengths an OVD suggests for image retrieval, while addressing and mitigating its inherent limitations.
>
> | Object-to-image area Ratio (%) | 0.2 | 0.4 | 0.8 | 1.0 | 1.5 | 2.0 |
> |-|-|-|-|-|-|-|
> | Object Recall               	| 0.68 | 0.83 | 0.83 | 0.85 | 0.90 | 0.87 |
>
> While there is a performance degradation in the extreme case of 0.2% the results show the high capability of owlv2 in discriminating even small objects.
>
> ### **Discussion on failure cases**
> Thank you. Beyond our discussion on limitations in Section 5 and Sections F and H in the Suppl. Material (see pointer to this section in L335-336, including OVD and clutter), we further noticed that our model is more biased toward objects and has shortcomings on faces, individuals, and landmarks. We will extend and add visual examples (omitted here due to formatting constraints) in the final version.
>
>
> ## Individual Concerns:
> ### **[W1] OVD on rare objects**
> Thank you for the comment. Please see our general response to this concern.
> We would like to clarify that in our retrieval task, the goal is not to classify the object’s category, but rather to obtain a discriminative representation suitable for similarity-based retrieval. This focus on representation, rather than recognition, helps mitigate challenges associated with rare or unseen objects. The OVD is used as object proposal (objectness)
> In fact, VoxDet dataset includes a wide range of fine-grained, long-tail, and open-vocabulary categories covering workshop and household hand tools, including cordless drill or Sextant (Nautical angle measuring instrument). Illustrative examples of such rare or hard-to-name categories, used by the benchmark, are shown in the top row of Figure 4.
>
> ### **[W2] In stage 1, ... How do you prompt the model …. Ensuring small objects are correctly cropped ..**
> During Stage-A training, we divide the dataset into query images and gallery images. Importantly, our method is entirely image-based, it does not use any text prompts in the task.
> Since our training data includes ground-truth bounding box annotations, we crop both the query and gallery images using these provided boxes (see L150–151). During gallery image encoding, we rely on an OVD to automatically generate bounding boxes for general objects in the gallery for each image (see L208).
> In gallery encoding stage (where there is no local annotation), we utilize OWLv2 as our OVD, to detect all objects in each image that then pass through MaO pipeline for encoding and aggregation to a single representation. For query images, which contain only a single object, the image is encoded and refined directly without the need for aggregation (see L173). We will rephrase to make this workflow clearer in the final version.
>
>
> ### **[W3] In stage 1, ... Small/Rare objects under-represented**
> Thank you for raising this important point. Our MaO approach specifically addresses the under-representation of small or rare objects, which is a known limitation of standard global representation methods. MaO improves their treatment in two key ways:
> Object cropping enhances effective “resolution”: By cropping individual objects, small objects are effectively assigned more input tokens in the ViT encoder (due to the fixed patching grid), leading to finer and more detailed representations. This advantage arises naturally from our detection-based design.
> Balanced aggregation: During feature pooling, all objects, regardless of size, contribute more uniformly weighted in the final representation.
> This mitigates the typical dominance of large objects and ensures that smaller ones (regardless of their rareness) are not overshadowed (see L59–61).
>
> ### **[W4] Why not fine-tune the vision encoder**
> Thank you for the suggestion. Our approach prioritizes obtaining a strong initial representation via average pooling, followed by refinement through object-mask-guided optimization, which is particularly beneficial for significance of small object representation.
> In our current design, Stage-B refines the global vector \hat{v}_c, based on a fixed initialization (with v_c) from Stage-A (as explained in L146). Allowing v_c​ (and the encoder) to be updated during this process would require a more complex optimization scheme, potentially involving alternate-optimization or EMA-based strategies, which would significantly increase computational overhead and complexity (maybe convergence issues) during training. We believe it represents an interesting direction for future work.
>
>
> ### **[W5] Comparing with …. Qwen (or Ferret)**
> Thank you for the suggestion. Current multimodal large language models (MLLMs) such as Qwen or Ferret are not well-suited for large-scale image retrieval. Their inference process typically relies on pairwise comparisons between the query image and each gallery image (e.g., prompting whether the object exists in a given image), which requires passing both images through the full MLLM pipeline. This makes the process computationally expensive and impractical for large-scale applications.
> However, in response to the reviewer’s comment, we also conducted a test using Qwen2.5-VL over 100 queries on a subset of the VoxDet dataset with a gallery of 200 images.
>
> | Method     	| Matching Type | Accuracy |
> |-|-|-|
> | Qwen2.5-VL 	| Pair Matching  | 19%  	|
> | MAO-DINOv2 	| Single Vector Retrieval | 85%  	|
>
> These results show that, beyond its impractical runtime, an MLLM like Qwen struggles with small-object retrieval, highlighting the advantages of dedicated retrieval models like MaO.
>
> ### **[W6] Computational complexity of the proposed method**
> We provide a detailed runtime analysis in Section D of the Supplementary Material, reporting both encoding and search times. Our method is compared against two representative SoTA methods: SuperGlobal (SG) for global matching, and AMES, which serves as an efficient local matching approach. This comparison highlights the efficiency and scalability of our method relative to these alternatives.
> At test time (i.e., from receiving a query image to producing ranked retrieval results), MaO requires 30 ms for query encoding (single object) and 0.03 ms for vectorized search (global matching) over 1,581 gallery images (VoxDet). In comparison:
> SG takes 65 ms for query encoding and 0.37 ms for search with global matching and re-ranking.
> AMES takes 65 ms for encoding and 2,286 ms retrieval due to the heavy local matching process.
> For offline encoding, both SG and AMES require 65 ms per image, while MaO takes 174 ms per image, in addition to ~0.5 seconds per image for the OVD step. However, this OVD cost can be significantly reduced through batching or optimized implementations (e.g., using ONNX), although such improvements were beyond the scope of this work. Notably, the entire MaO encoding pipeline, including the OVD step, is highly parallelizable due to its data-parallel nature, offering further opportunities for encoding speed-ups.
>
> ### **[W7] Failure cases**
> Please see our general response

---

> > ### Author Response · Authors · 2025-08-06
> >
> > Thank you for your time and effort reviewing our paper and rebuttal.
> >
> > We hope our response and additional experiments have fully addressed your concerns. If any remain, we would be happy to provide further clarification.

---

### Official Review · Reviewer_23tf · 2025-07-17

**Clarity:** 2
**Significance:** 3
**Originality:** 2
**Rating:** 4
**Confidence:** 5

**Summary:**

This paper introduces Multi-object Attention Optimization (MaO), a novel framework for Small Object Image Retrieval (SoIR) in cluttered scenes. MaO proposes a two-stage approach: multi-object fine-tuning followed by attention-based refinement using explainability maps and object masks to create a unified image descriptor. The authors also introduce new benchmarks, featuring smaller objects and higher clutter. Experiments demonstrate MaO's superior performance on these benchmarks compared to existing methods and foundation models.

**Questions:**

[Q1] Given that the paper primarily focuses on the effectiveness of MaO for small objects, could the authors provide an explicit analysis of MaO's performance on large objects or its applicability and effectiveness for general image content-based retrieval beyond object-centric queries?

[Q2] could the authors elaborate on the potential impact of significant failures in object detection or localization by the OVD on MaO's "in-the-wild" retrieval performance? Particularly in highly cluttered scenes where such failures might be more prevalent.

[Q3]  Could the authors provide a more in-depth explanation of how explainability maps are precisely computed and optimized within LeGrad, specifically addressing the mechanism when multiple objects contribute to a single global token?

[Q4] Could the authors provide a more detailed comparison highlighting MaO's specific advantages over  $\alpha$-CLIP and other mask-based methods in handling multi-instance, cluttered scenes, beyond just the mAP difference?

[Q5] Given that Stage B optimization takes 0.03 seconds per-object, could the authors provide a more detailed explanation of how this contributes to the total image encoding time, and clarify the total online inference time for a single query, including OVD detection?

**Ethical Concerns:**

["NO or VERY MINOR ethics concerns only"]

**Final Justification:**

Thank you to the reviewers for their time and effort. I choose keep my rating.

**Limitations:**

yes

**Quality:**

2

**Strengths And Weaknesses:**

[S1] MaO offers an innovative solution to the challenging and practical problem of retrieving small objects in cluttered images, a task where traditional methods often fall short.

[S2] MaO consistently outperforms state-of-the-art instance retrieval methods and foundation models across various settings, showing robust improvements for small object retrieval.

[S3] The framework's ability to integrate with different vision backbones and perform well in both zero-shot and fine-tuned scenarios highlights its adaptability and potential for broad applicability.



[W1] While highly effective for small objects, the paper's explicit analysis of MaO's performance on large objects or general image content-based retrieval is limited.

[W2] MaO's "in-the-wild" performance depends on the accuracy of the OVD. While the method's refinement step mitigates some OVD imperfections, significant failures in object detection or localization by the OVD could impact retrieval performance, particularly in highly cluttered scenes.

[W3] Although LeGrad is a key component, the main paper could benefit from a more in-depth explanation of how explainability maps are precisely computed and optimized in the context of multiple objects contributing to a single global token.

---

> ### Author Rebuttal · Authors · 2025-07-30
>
> Thank you for your valuable feedback and positive assessment, recognizing our approach as an **“innovative solution, highlights adaptability and potential for broad applicability”** and noting it **“consistently outperforms SoTA.”**
>
> ## General Concerns:
> ### **“The MaO pipeline is critically dependent on an upstream OVD.**
>
> This raised concern is an inherent limitation of any modular, detection-first retrieval pipeline, and we appreciate the opportunity to address it in more depth.
>
>
> 1. Dependence on an OVD is a standard practice across several domains.
> Our design follows a well-established paradigm that is widely adopted in various methods e.g., Grounded-SAM (combining Grounding DINO OVD), Open3DIS (CVPR 2024) that performs open-vocabulary 3D instance segmentation, and TROY-VIS (WACV 2025) that uses an existing OVD for video instance tracking.
> Thus, the use of an OVD as a building block is not unique to our method, but consistent with many best practices in the field. Novel OVD methods can operate even under extreme object size and clutter conditions (see table below), and we leverage this in our approach.
>
>
> 2. No need for object classification, just object proposals
> We use OVD in the “objectness” mode (L207-208) where the model is used as an object proposal and needs to binary discriminate an image area as object or background.
>
> 3. In practice, the impact is limited and measurable.
> To further address this concern (that was discussed in Section 5), we conducted an empirical evaluation of our used OVD model on two GT-annotated benchmarks, and reported in our  Suppl. Material (SM) in Section F (as referred in L350 in the main paper). The results show that even under imperfect detection (partial object detection) our method continues to outperform existing approaches.
> In summary, we believe our method effectively harnesses the strengths an OVD suggests for image retrieval, while addressing and mitigating its inherent limitations.
>
> | Object-to-image area Ratio (%) | 0.2 | 0.4 | 0.8 | 1.0 | 1.5 | 2.0 |
> |-|-|-|-|-|-|-|
> | Object Recall               	| 0.68 | 0.83 | 0.83 | 0.85 | 0.90 | 0.87 |
>
> While there is a performance degradation in the extreme case of 0.2% the results show the high capability of owlv2 in discriminating even small objects.
>
> ### **Discussion on failure cases**
> Thank you. Beyond our discussion on limitations in Section 5 and Sections F and H in the Suppl. Material (see pointer to this section in L335-336, including OVD and clutter), we further noticed that our model is more biased toward objects and has shortcomings on faces, individuals, and landmarks. We will extend and add visual examples (omitted here due to formatting constraints) in the final version.
>
> ## Individual Concerns:
> ### **[W1,Q1] Mao Performance on large and general image content based retrieval**
> Thank you for this suggestion. In the SM (Section C), we report results on ILIAS dataset, which includes a mix of small, medium, and large objects. To further address this concern, we conducted an additional evaluation on PODS dataset [1], which has an average object-to-image area-ratio of 20%. As shown in the table below, MaO performs robustly on these larger instances, with no degradation in performance.
> Results (with finetuned models) on non-small objects, PODS Dataset:
> | Method| mAP|
> |-|-|
> | CLIP| 0.341 |
> | DINOv2| 0.332 |
> | MaO-CLIP| 0.368 |
> | MaO-DINOv2| 0.393 |
>
> [1] Sundaram et al,, Personalized representation from personalized
> Generation, https://arxiv.org/abs/2412.16156
>
> ### **[W2]: Performance of the OVD in cluttered scenes**
>
> Please see our answer on OVD limitations in the general response.
> More specifically, we provide a breakdown of the used OVD (on VoxDet) according to several clutter levels. The OVD maintains strong performance even in extreme cases where 12-24 objects are detected in the image.
>  | # Objects | 12   | 16   | 20   | 24   |
> |-|-|-|-|-|
> | Recall    | 0.80 | 0.95 | 0.95 | 0.89 |
>
>
> ### **[Q3,W3] More in-depth explanation on optimization**
> Thank you for the suggestion. We have provided a preliminary explanation of LeGRAD and the explainability map in Section A of SM. Essentially, we optimize a single vector with respect to K different image crops (K objects), while backpropagating gradients only through this single (unified) vector (as illustrated in Fig.2). This process ensures that the final vector effectively represents all objects present We will expand this discussion in the final version, either in the main paper or the Supplement.
>
>
> ### **[Q4] MaO specific advantage over $\alpha$-CLIP**
> To highlight the specific advantages of MaO over $\alpha$-Clip, which also employs mask-based representations, we conducted a focused experiment. We selected 200 multi-object images from the VoxDet dataset and computed their representations using both $\alpha$-CLIP (with masks over multiple objects) and MaO-CLIP. We measured the difference in distance between each object’s representation and two reference sets: (i) its corresponding source image containing the object, and (ii) a set of negative images that do not contain the object (higher values indicate better performance). In the MaO-CLIP space, this measure was on average twice as large as in $\alpha$-CLIP, indicating that MaO yields more distinct and better-separated embeddings.
> This implies superior aggregation and encoding, in our MaO approach as opposed to $\alpha$-CLIP, which was primarily designed for single, often large, objects.
>
> ### **[Q5] Total image encoding and inference time**
> Please note that we provide a runtime analysis in Section D of the Suppl Material.
> In response to the reviewer’s concern, we distinguish between two stages of computation:
> (1) Offline encoding time, which includes OVD detection and feature extraction for the gallery images,
> (2) Online query encoding and global search time, which covers real-time inference from a query image.
> At inference time, defined as the duration from receiving a query image to producing the ranked retrieval results, the process takes 30 ms and 0.03 ms for vectorized search and ranking over 1,581 gallery images (VoxDet). We employ FAISS for efficient vector-based retrieval.
> The OVD component, used only during gallery preprocessing (offline), adds approximately 0.5 seconds per image. This step does not affect online latency and can be significantly reduced with batching or optimized implementations such as ONNX, though this was beyond the scope of our current work. It is worth noting that the entire encoding pipeline, including OVD, is highly parallelizable due to its data-parallel nature.

---

> > ### Author Response · Authors · 2025-08-06
> >
> > Thank you for your time and effort and for recognizing our approach as an “innovative solution” that “highlights adaptability and potential for broad applicability.”
> >
> > We hope our response and additional experiments have fully addressed your questions. If there is anything further we can clarify, we would be happy to do so.

---

### Note · Authors · 2025-08-14

We thank the reviewers for their constructive remarks.
Our paper introduces three main contributions: **(1)** Identifying a critical shortcoming of existing retrieval methods when handling small objects, in cluttered scenes. **(2)** Introducing a new benchmark that specifically evaluates this phenomenon and establishes a foundation for future research.
**(3)** Proposing a novel method that handles images containing small objects even under clutter, while using a single representation, to reduce memory footprint for large-scale image retrieval.

The reviewers raised three main concerns:

**(a) Dependence on OVD:** We argued that two-stage designs are common in computer vision, as seen e.g. in SAM that uses an OVD as its first stage, or in many object detectors that rely on object proposal methods (more examples are provided in the rebuttal). In our case, the OVD is used purely as an objectness (object proposal) stage. We further presented an analysis showing that modern OVDs (e.g., OWLv2) are capable of detecting small and even unusual objects effectively.
**(b) Runtime evaluation:** We referred to our supplementary material, which shows that our approach incurs minimal encoding overhead while enabling fast and efficient online search. Additional details were provided in the rebuttal.
** (c) Failure case discussion:** Addressed in our rebuttal with clarifications and further analysis available in our suppl. material

On most of these issues, we referred the reviewers to our supplementary material, as our paper presents several substantial contributions and there was insufficient space to include all details in the main paper.

All reviewers acknowledged our first two contributions. While Reviewer R-23tf explicitly described our method as an “innovative solution”, and R-Ey8L highlighted the two-stage design as a strength, R-1h7Q raised a concern about the novelty, which we respectfully addressed in both the rebuttal and post-rebuttal responses.

Finally, we also provided additional experimental evidence on the validity of our method, including comparisons involving MLLMs (Qwen), and performance on large-objects.

---

### Decision · Program_Chairs · 2025-09-17

**Decision:**

Accept (poster)

**Comment:**

This paper received initially 1 Borderline Accept and 2 Borderline Reject scores. While reviewers praise the interesting/important problem in a relatively unexplored area, new benchmarks and extensive experiments, there were concerns around limited novelty and the technique’s dependence on an open-vocabulary detector. The rebuttal was regarded as helpful by all reviewers, clarified most of the concerns, leading one reviewer to upgrade their rating from Borderline Reject to Borderline Accept. All ratings in the end are borderline, but leaning positive. The main remaining concern is the somewhat limited novelty of the paper. However, the authors discuss this in the rebuttal, and the AC considers it in the end as satisfactory, especially taking into account that this is an area not very actively explored and this work can help push forward this important problem. After careful consideration, the AC is supportive of accepting the paper into NeurIPS, given all its positive aspects and the extensive additional information provided in the rebuttal. We require that the clarifications and all relevant information provided in the rebuttal be incorporated in the camera-ready version.